# Regional and global impact of $CO_2$ uptake in the Benguela Upwelling System through preformed nutrients

Claire Siddiqui [1] ✉, Tim Rixen [1,2], Niko Lahajnar [2], Anja K. Van der Plas [3], Deon C. Louw [4], Tarron Lamont[5,6,7] & Keshnee Pillay [5]

Eastern Boundary Upwelling Systems (EBUS) are highly productive ecosystems. However, being poorly sampled and represented in global models, their role as atmospheric $CO_2$ sources and sinks remains elusive. In this work, we present a compilation of shipboard measurements over the past two decades from the Benguela Upwelling System (BUS) in the southeast Atlantic Ocean. Here, the warming effect of upwelled waters increases $CO_2$ partial pressure ($pCO_2$) and outgassing in the entire system, but is exceeded in the south through biologically-mediated $CO_2$ uptake through biologically unused, so-called preformed nutrients supplied from the Southern Ocean. Vice versa, inefficient nutrient utilization leads to preformed nutrient formation, increasing $pCO_2$ and counteracting human-induced $CO_2$ invasion in the Southern Ocean. However, preformed nutrient utilization in the BUS compensates with ~22–75 Tg C year$^{-1}$ for 20–68% of estimated natural $CO_2$ outgassing in the Southern Ocean's Atlantic sector (~ 110 Tg C year$^{-1}$), implying the need to better resolve global change impacts on the BUS to understand the ocean's role as future sink for anthropogenic $CO_2$.

Eastern Boundary Upwelling Systems (EBUS) are among the most productive regions in the ocean and contribute 11% to global new production, which refers to biomass largely produced on the basis of nutrients introduced via upwelling and vertical mixing from the deep dark ocean into surface waters[1–3]. The associated assimilation of $CO_2$ through the generation of biomass and its transfer to the deep sea is an integral part of the biological carbon pump[4,5] which reduces atmospheric $CO_2$ concentrations through the storage of $CO_2$ as dissolved inorganic carbon (DIC) in the deep ocean. Even though it is widely assumed that the biological carbon pump responds to climate change, the magnitude and even the direction of change is still unknown[6–8]. However, the amount of DIC kept by the biological

carbon pump is assumed to be linearly related to the inventory of regenerated nutrients, which are released during the remineralization of biomass in the deep ocean[9]. They stand in contrast to biologically unused so-called preformed nutrients, whose formation represents a leakage through which the biological carbon pump loses $CO_2$ and makes e.g. the Southern Ocean south of 44°S a natural $CO_2$ source to the atmosphere[10,11]. This leakage evolves when nutrients that upwell along with DIC are not fully utilized by biological production due to light and iron limitation[12–14]. Nowadays, rising $CO_2$ concentrations in the atmosphere reverse the air-sea fluxes of $CO_2$ and have converted the Southern Ocean into a key site for anthropogenic $CO_2$ uptake with a current estimate of approximately 740 ± 290 Tg C year$^{-1}$ [15,16]. Due to

[1]Leibniz Centre for Tropical Marine Research - ZMT, Fahrenheitstrasse 6, 28359 Bremen, Germany. [2]Institute of Geology, Universität Hamburg, Bundesstrasse 55, 20146 Hamburg, Germany. [3]National Marine Information and Research Centre, PO Box 912, Swakopmund 13001, Namibia. [4]Debmarine Namibia, 10 Dr Frans Indongo Street, Windhoek 10005, Namibia. [5]Oceans & Coasts Research Branch, Department of Environment, Forestry and Fisheries, PO Box 52126, Victoria & Alfred Waterfront, Cape Town 8000, South Africa. [6]Marine Research Institute & Department of Oceanography, University of Cape Town, Rondebosch, South Africa. [7]Bayworld Centre for Research & Education, 5 Riesling Road, Constantia, Cape Town 7806, South Africa. ✉e-mail: claire.siddiqui@leibniz-zmt.de

the counteracting effect of natural $CO_2$ released from the biological carbon pump[13] of about 400 ±180 Tg C year$^{-1}$ [15,16], this results in a $CO_2$ invasion of 340 ±110 Tg C year$^{-1}$ between 1990 and 2009[16–18]. However, approximately 27.5% (-110 Tg C year$^{-1}$) of the natural $CO_2$ release from the biological carbon pump occurs within the Atlantic sector of the Southern Ocean between 44° and 58°S[15,16], while the resulting preformed nutrients are transported northwards and subducted beneath warmer and lighter subtropical water masses[19–21]. Loaded with preformed nutrients, these mode waters support primary productivity in upwelling systems at lower latitudes and thus could potentially restore the $CO_2$ uptake efficiency of the biological carbon pump. Although EBUS could therefore act as regional $CO_2$ sinks through the utilization of preformed nutrients and low sea water temperatures which increase the solubility of $CO_2$[22–24], global models suggest that upwelling systems (in particular ones at lower latitudes) act as net $CO_2$ sources to the atmosphere[25–27]. This was also found to be the case in a modelling study of the Benguela Upwelling System (BUS), which is located in the southeast Atlantic Ocean (Fig. 1) and considered the most productive of all EBUS[28]. The study however suffered from a model that poorly represented the BUS, and an unclearly defined upwelling region[27]. Other studies within the BUS have not resolved these issues as indicated by estimated $CO_2$ fluxes ranging from −5.1 to 1.54 Tg C year$^{-1}$ [25,26,29], with opposing air-sea fluxes[30] noted between the northern (11.5 Tg C year$^{-1}$)[31] and southern (−1.4 to −2.8 Tg C year$^{-1}$)[31,32] upwelling area. It thus remains elusive whether the BUS is a net $CO_2$ sink or $CO_2$ source to the atmosphere. Similar to the Southern Ocean, the BUS also suffers from a sparsity of data, and model simulations and evaluations[26,27] were constrained by pCO$_2$ data (partial pressure of $CO_2$) from the Surface Ocean $CO_2$ Atlas (SOCAT)[33]. So far this data product misses coverage particularly over the northern BUS shelf region, thus curtailing estimates of air-sea gas exchange for the BUS (Fig. 1).

Here, we address this problem by presenting a compilation of shipboard pCO$_2$ data for the BUS from the extended SOCAT v2020[33] data base that includes data gathered from 14 cruises throughout the BUS from 2008 to 2019 (see Supplementary Table 1). This revised data set allows for a seasonal-based examination of regional air-sea fluxes of $CO_2$ across the BUS and underlying mechanisms affecting these fluxes, which indicate that the biological carbon pump in the BUS serves as a globally significant $CO_2$ sink.

## Results and discussion
### Sea surface pCO$_2$ characteristics
In order to overcome the problem of unclearly defined upwelling regions that led to strong discrepancies in the definition of the BUSs offshore boundary[26,31,34], we specified the upwelling zone's spatial extent by considering the average cross-shelf distribution of sea surface pCO$_2$ (Fig. 2a, b) for the northern and southern subsystems (NBUS, SBUS). The latitudinal boundary between the NBUS and SBUS is formed by the Lüderitz cell (-26°S)[35–37] within the Lüderitz upwelling region (24°S–28°S)[38,39] that is subject to perennial coastal upwelling. In general, upwelling systems exhibit highest pCO$_2$ in the nearshore region due to the upwelling of carbon-rich waters, and a decreasing trend offshore due to degassing and the biologically-mediated carbon uptake within the offshore-advecting upwelled water[34,40,41]. By assuming that a decrease in pCO$_2$ variability marks a decreasing influence of upwelling on pCO$_2$, we determined the upwelling zone's boundary as the distance from shore when the standard deviation (s.d.) decreases persistently to below ± 30 µatm in both subsystems.

In the NBUS, this results in an upwelling boundary at 340 km offshore, which lies within the lower range of other studies that considered an offshore extension from 300 to 800 km[27,28,42,43]. In the SBUS, the decreasing variability in pCO$_2$ suggests a closer boundary at 200 km distance to shore. In comparison to the NBUS, the much lower intensity of upwelling observed in the SBUS[44,45] may explain the

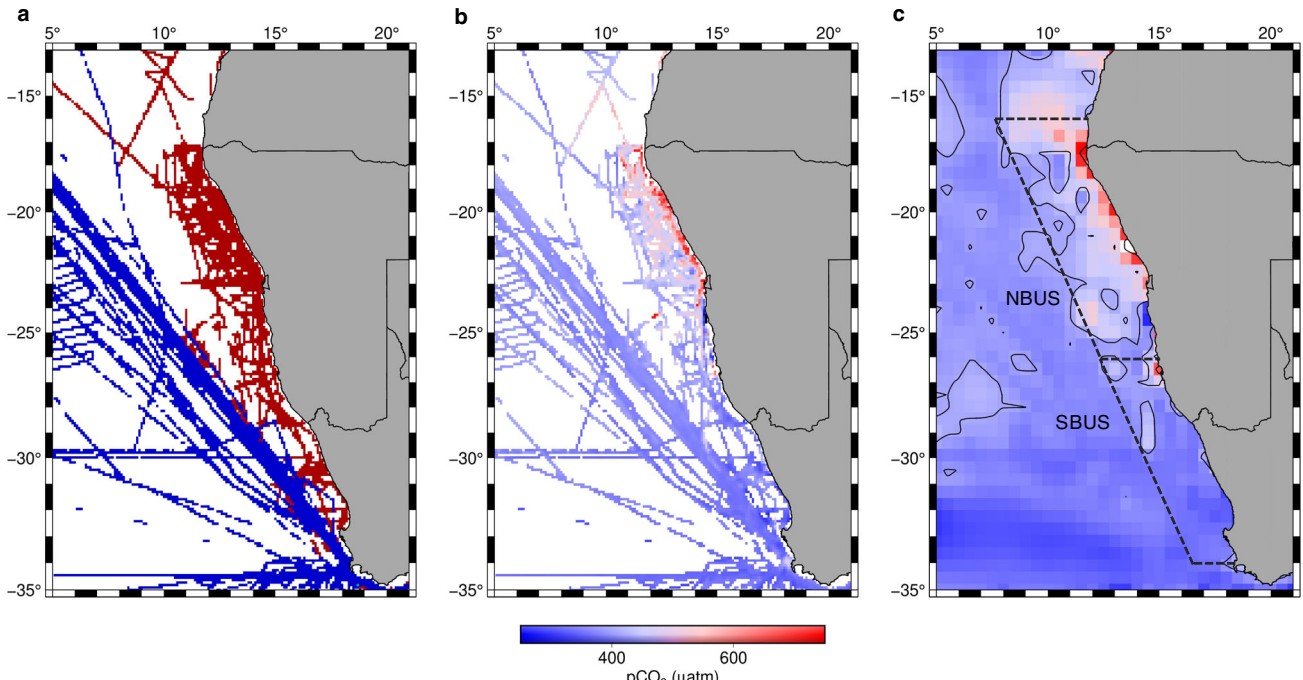

**Fig. 1 | Maps of pCO$_2$ measurements of research vessel cruises within the Benguela Upwelling System used in this study. a** Underlying cruise tracks of the 13 cruises (red) and those embedded in SOCAT v2020[33] (blue). **b** Recorded pCO$_2$ measurements (in µatm), normalized to the reference year 2020. **c** Normalized pCO$_2$ measurements (in µatm) interpolated on a 0.5° grid. Dashed lines mark the area of the northern and southern Benguela Upwelling System (NBUS, SBUS), while contour lines represent the atmospheric pCO$_2$ of the reference year 2020 based on Mauna Loa records (414 µatm). The interpolation was performed after the minimum curvature interpolation method[105] implemented in the Generic Mapping Tools (GMT). Country outlines sourced from the GSHHG dataset[106] and were plotted with GMT.

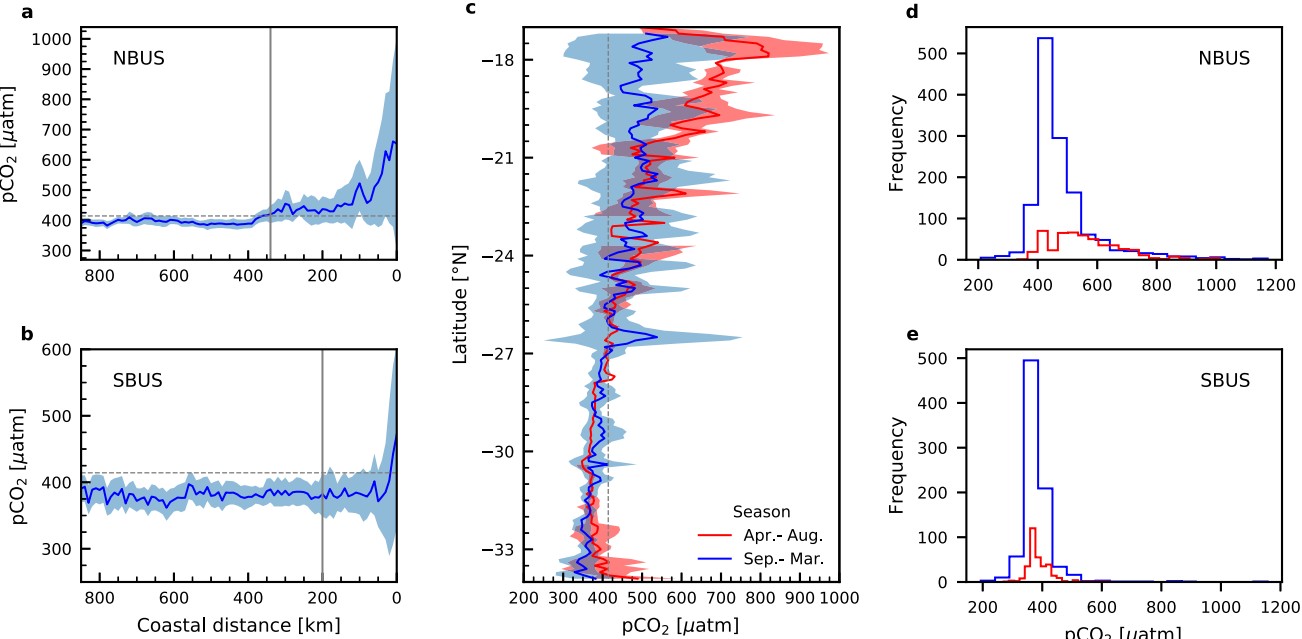

**Fig. 2 | Spatial and temporal variability of sea surface pCO₂. a** Average cross-shelf distribution of all pCO₂ measurements (in μatm) of the northern part of the Benguela Upwelling System (NBUS) with distance to the coast (in km). The grey line marks the offshore boundary of the upwelling zone. Values were normalized to the reference year 2020, and averaged over intervals of 10 km coastal distance. **b** As in a, except for the southern part (SBUS). **c** Average latitudinal pCO₂ concentrations across the NBUS and SBUS regions, for austral spring and summer (blue) and austral autumn and winter (red). **d** Histogram of pCO₂ within the NBUS for both seasons. **e** As in **d**, except for the SBUS. The dashed grey line in subplot **a**, **b** and **c** represents the atmospheric pCO₂ of the reference year 2020 based on Mauna Loa records (414 μatm), while the shaded areas represent the standard deviation.

discrepancy observed in the offshore extent of elevated $pCO_2$ values between the subsystems. Considering additionally the latitudinal extent of the SBUS (26°S to 34°S) and NBUS (16°S to 26°S) results in upwelling areas of 177.600 km² and 377.400 km², respectively.

Relative to the mean atmospheric $pCO_2$ of our reference year 2020 (~414 μatm), the annual mean $pCO_2$s of 492.3 ± 115.82 μatm for the NBUS and 383.9 ± 53.73 μatm for the SBUS reveal the opposing character of the two subsystems as a regional $CO_2$ source and sink, respectively. The uncertainty in the mean annual $pCO_2$ could thereby be attributed to the variability of $pCO_2$ in close proximity to the coast due to the impact of upwelling and the seasonality in upwelling intensities. To estimate the latter, $pCO_2$ data were averaged across the mean offshore extent and latitudes of the upwelling areas for the austral spring and summer (September-March), and austral autumn and winter seasons (April-August, Fig. 2c). The resulting plot of $pCO_2$ variability with latitude indicates a seasonal influence in the northern region between ~17°S and ~21°S which diminishes towards the south. The observed seasonality with enhanced $pCO_2$ between April and August in the north corresponds to seasonal variations in upwelling intensities, which for this region are strongest during this time of the year[46]. However, when averaged across the whole NBUS and considering the standard deviation (Fig. 2c), the ~17% difference in seasonal means (481.0 ± 117 μatm, September-March versus 560.7 ± 66 μatm, April-August) imply that the upwelling-related seasonality is only weakly pronounced. In the SBUS, seasonal mean $pCO_2$s of 381.7 ± 36 μatm (September-March) and 388.5 ± 44 μatm (April-August) do not reflect the seasonality of upwelling intensities, which are strongest during the austral spring and summer (September-March)[47]. This could be attributed to the balancing between initial outgassing and biologically-mediated $CO_2$ uptake under intensified upwelling conditions, and minor effects of vertical water mass transports and biology on the $pCO_2$ during the non-upwelling season. Although our data mirrors the upwelling-related variability in $pCO_2$ for each subsystem during both seasons (Fig. 2d, e), there are more measurements

available for the austral spring and summer season (September-March) (see also Supplementary Fig. 1 and 2). Hence, assumptions on seasonal differences and annual estimates for a given subsystem should be treated carefully. Overall, in line with annual means, the seasonal $pCO_2$ estimates display the opposing behaviour of the NBUS and SBUS as a regional $CO_2$ source and sink, respectively.

## Air-sea CO₂ flux estimates

In contrast to $pCO_2$, air-sea $CO_2$ fluxes reveal a pronounced seasonality in both systems as being more than twice as high during upwelling than during non-upwelling seasons. The intensification of wind during the upwelling season is considered the primary driver of this difference which, given the opposite signs of the flux in the two regions, strengthens the $CO_2$ source and sink functions in the NBUS and SBUS, respectively (Supplementary Table 2). Integrating the annual mean fluxes over the upwelling area and considering their uncertainties results in a $CO_2$ emission of 15.64 (−2.95–73.16) Tg C year⁻¹ in the NBUS and a $CO_2$ uptake of −2.94 (−4.5–3.58) Tg C year⁻¹ in the SBUS. Taking additionally the different gas transfer velocity parameterizations into account (see Methods section, Supplementary Table 3), annual mean $CO_2$ fluxes can increase in both subsystems by ~71% to 26.69 Tg C year⁻¹ (NBUS) and −5.03 Tg C year⁻¹ (SBUS), underlining the profound difference in the sink and source character of the SBUS and NBUS, respectively. However, in comparison to our initial area-integrated $CO_2$ fluxes given in ref. 31. for the NBUS (11.5 Tg C year⁻¹) and SBUS (−1.4 Tg C year⁻¹), our respective estimates of 15.64 (NBUS) and −2.94 (SBUS) Tg C year⁻¹ are about 40 and 110% higher, mainly due to the use of a smaller area in the ref. 31. study, but potentially also due to less shipboard measurements that were used to calculate the $CO_2$ fluxes. Similar to these data-based estimates, modelled carbon fluxes also indicate a $CO_2$ sink region in the south and source region in the north[27], while modelled $CO_2$ fluxes from the BUS between 18°S to 28°S amount to ~24 Tg C year⁻¹ on average over the time period between 1982 and 2015. Comparing our measurements with modelled data by

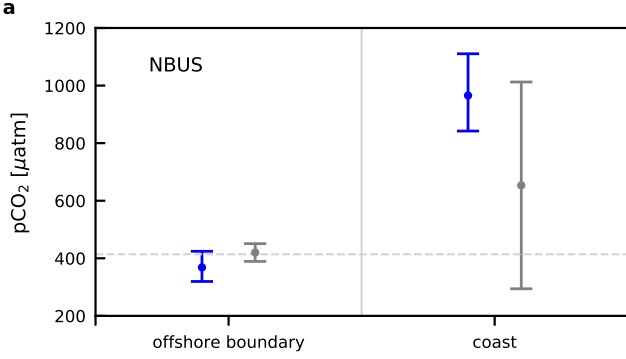

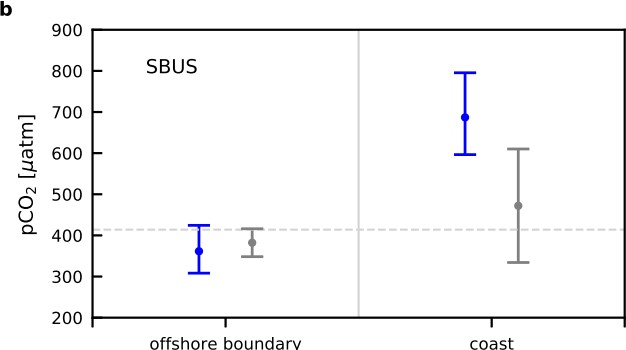

**Fig. 3 | Thermal and biological controls on sea surface $pCO_2$ based on CO2SYS simulations.** Measured (grey) and modelled (blue) sea surface $pCO_2$ (in µatm) during coastal upwelling and nitrate (N) consumption through biologically-mediated $CO_2$ uptake at the offshore boundary, based on CO2SYS[96,97] calculations using source water mass characteristics (Supplementary Table 4) for the **a** northern (NBUS) and **b** southern Benguela Upwelling System (SBUS). The simulated $pCO_2$ at the offshore boundary includes the effect of surface warming on $pCO_2$. The grey dashed line represents the atmospheric $pCO_2$ of the reference year 2020 based on Mauna Loa records (414 µatm). Sea surface $pCO_2$ concentrations below (above) the atmospheric level indicate a source (sink) of atmospheric $CO_2$. The uncertainties in measured $pCO_2$ are presented as the standard deviation, whereas uncertainties in the modelled $pCO_2$ are based on the standard error of the average source water mass characteristics.

recalculating $CO_2$ fluxes for the same region with the same offshore boundary at 800 km, results in a weaker annual $CO_2$ source into the atmosphere of ~8 Tg C year$^{-1}$. This discrepancy seems to be caused mainly by a poleward misplacement of the outgassing cell in the applied model framework[27], which underestimates the $CO_2$ sink behaviour of the SBUS.

## Nutrients as a driver of regional variability in sea surface $pCO_2$

In order to differentiate between effects of surface-warming and the biologically-mediated drawdown of $CO_2$ on the air-sea gas exchange, we followed a commonly applied bottom-up approach to estimate new and export production, assuming nutrients of upwelled source waters to be consumed at the surface and transformed into organic matter (new production) that is subsequently exported out of the euphotic zone (export production)[1,3,48].

Before we used this approach, it was validated in two steps. In a first step we assumed surface-warming and the biologically-mediated drawdown in nearshore regions to largely be negligible and the high measured $pCO_2$ to be an immediate consequence of the cold, DIC and nutrient-enriched source waters that are introduced into the surface layer. Hence, we first calculated the potential $pCO_2$ in freshly upwelled waters based on temperature, salinity, as well as TA, DIC and nutrient concentrations of the upwelling source water masses (ESACW and SACW, Supplementary Table 4) and compared the resulting $pCO_2$ with those measured in the nearshore regions.

In a second step, surface warming and effects of nutrient utilization on $pCO_2$ in surface waters were estimated by taking into account that phytoplankton consumes upwelled nutrients to fix DIC into biomass, as also indicated by data showing nutrient depletion of offshore flowing upwelled waters at distances of about 180-200 km to the coast[49,50]. The Redfield ratio (106:16) is, in turn, used to translate nutrient utilization into DIC consumption and the associated release of total alkalinity (see e.g[1,48].). By subtracting the amount of DIC which is transformed into organic matter from the original source water DIC concentration, and adding the released TA to the original source water TA, we can calculate the $pCO_2$ in upwelled water after upwelled nutrients have been consumed and exported. During the offshore flow, upwelled waters warm up as indicated by the difference between temperatures of the source waters and the sea's surface temperature. Hence, by using sea surface temperatures and salinities instead of those of the source water, we can further consider the warming of upwelled waters and its effect on $pCO_2$.

The exercise implies, in line with our data (Fig. 2a, b), a high $pCO_2$ in the freshly upwelled water near the coast that decreases towards the outer boundaries of the subsystems where nutrients are consumed (Fig. 3). The mean calculated $pCO_2$ in the freshly upwelled water (first step: NBUS: 842–1110 µatm, SBUS: 596–795 µatm) falls within the upper range of the nearshore recorded $pCO_2$ (NBUS: 294–1012 µatm, SBUS: 334–610 µatm). This implies that biologically-mediated drawdown of $CO_2$ occurs simultaneously with coastal upwelling as indicated by e.g., high satellite-derived chlorophyll concentrations at a narrow belt along the nearshore region[51–53]. At the outer boundaries where nutrients are already consumed, measured values fall, in turn, well within the ranges of calculated ones (second step: NBUS: 319–423 µatm, SBUS: 310–426 µatm). The correspondence between the measured and calculated $pCO_2$ within both subsystems (Fig. 3) places confidence in our bottom-up approach and validates its use for estimating effects of surface warming and the biologically-mediated drawdown of $CO_2$ on the $pCO_2$ in surface waters. To further resolve the latter effects on a spatio-temporal scale, we reconstructed the non-thermally controlled $pCO_2$. Therefore, we calculated DIC based on our $pCO_2$ climatology which includes annual and seasonal gridded fields on $pCO_2$, SST and SSS, as well as the SSS and TA correlation (see Methods for a detailed description). To eliminate the surface warming effect, $pCO_2$ was recalculated by using DIC and TA but instead of using SST and SSS, source water salinities and temperatures were used. Hence, the non-thermally controlled $pCO_2$ indicates the $pCO_2$ which one would have expected in case upwelling waters would not have been warmed at the surface.

As a result, the non-thermally controlled $pCO_2$ as depicted in Fig. 4 mirrors spatial and temporal trends seen in the measured $pCO_2$ (Fig. 2), but remains below the measured $pCO_2$. Hereby, the non-thermally controlled $pCO_2$ falls below the atmospheric $pCO_2$ at the outer boundaries, indicating that the consumption of upwelled nutrients would have turned both subsystems into $CO_2$ sinks to the atmosphere if not overcompensated by surface warming in the NBUS (Fig. 4a, b). Even though the latitudinal averages express this during the austral spring and summer season (September-March), the austral autumn and winter season (April-August) does not reflect this trend (Fig. 4c, d) due to sample biases. As discussed before, less measurements were available from the offshore region from this time period so that the available nearshore data dominate the mean. At these sites, the biological carbon pump is still quite inefficient as upwelled nutrients have not fully been consumed, causing the regenerated DIC which upwells along with the nutrients to increase $pCO_2$ in surface waters (Fig. 4a). In this regard, the availability of nutrients, in addition to changes in the Redfield carbon to nutrient ratio, largely control the potential biologically-mediated $CO_2$ uptake. Since in the BUS, the Redfield carbon to nutrient ratio is assumed to be constant[49,54–56], its impact on biologically-mediated $CO_2$ uptake is neglected in the following

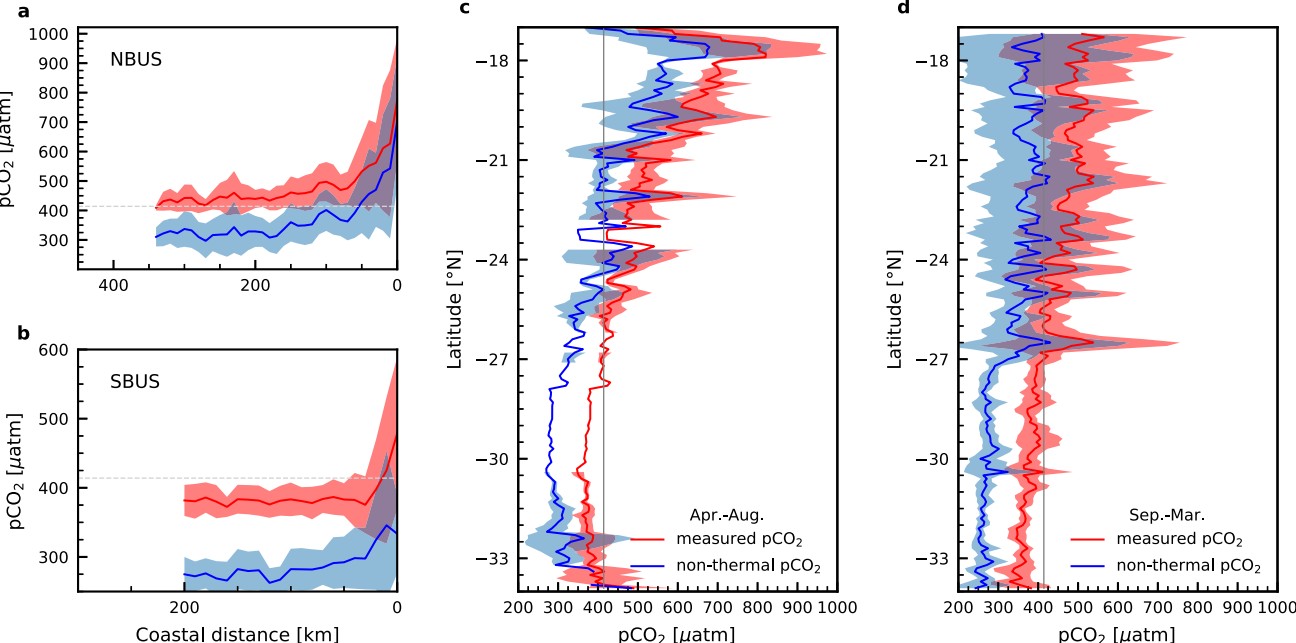

**Fig. 4 | Spatio-temporal variability of the non-thermally controlled sea surface pCO₂ based on CO2SYS simulations.** Average cross-shelf distribution of measured (red) and non-thermally controlled pCO₂ (blue) (in μatm) with distance to the coast (in km) based on CO2SYS[96,97] calculations using sea surface data and source water mass characteristics (Supplementary Table 4) for the **a** northern (NBUS) and **b** southern Benguela Upwelling System (SBUS). Values were averaged over intervals of 10 km coastal distance. **c** Average latitudinal pCO₂ concentrations (measured and non-thermally controlled) across the NBUS and SBUS regions during austral spring and summer. **d** As in **c**, except for austral autumn and winter. The grey lines in each subplot represent the atmospheric pCO₂ of the reference year 2020 based on Mauna Loa records (414 μatm). Sea surface pCO₂ concentrations below (above) the atmospheric level indicate a source (sink) of atmospheric CO₂. The uncertainties in measured and non-thermally controlled pCO₂ are presented as the standard deviation.

discussion. The drawdown of CO₂ by the biological consumption of nutrients plays, in turn, different roles in the marine carbon cycle: The regenerated nutrient consumption balances the input of regenerated CO₂ from below the euphotic zone whereas the utilization of preformed nutrients compensates for CO₂ outgassing during their formation at higher latitudes. Hence, changes in the regenerated nutrient consumption do not affect the net flux of CO₂ across the air-sea interface, because of associated variations in the supply of regenerated nutrients and CO₂, while the utilization of preformed nutrients can affect the net CO₂ uptake by compensating CO₂ losses from the Southern Ocean[57–59] (see Methods). Hence, it is the relative strength of the warming-driven increase of pCO₂ and consumption of preformed nutrients that finally controls the regional CO₂ sink and source functions of the SBUS and NBUS.

To assess the CO₂ uptake by the utilization of preformed nitrate ($N_{pref}$) in the SBUS and NBUS, we followed the bottom-up principle to calculate new/export production rates as mentioned earlier[1,60]. However, instead of using the total amount of nitrate, we only consider the preformed nitrate concentration within the upwelling source waters, which have been compiled from different cruises (see Supplementary Table 4). Hence, $N_{pref}$ of 6.44 ± 0.61 and 7.44 ± 0.36 μmol kg⁻¹, with an upwelling volume of 0.9 Sverdrup (Sv) for the NBUS[45,60] and 0.4 Sv for the SBUS[45], amount to new production rates driven by the utilization of $N_{pref}$ of 14.5 ± 1.4 and 7.5 ± 0.4 Tg C year⁻¹, respectively.

Alternatively, this part of the new production can also be estimated by using previously published new production rates[30,31,48] and the contribution of preformed nutrients to the total nutrient concentrations as derived from our water mass characteristics (see Supplementary Table 4). The published new production rates fall within a comparatively wide range of 68–245 T C year⁻¹ for the NBUS and 13.5–42 Tg C year⁻¹ for the SBUS[30,31,48], which in sum covers published new production rates of 241 Tg C year⁻¹ derived for the entire BUS from 16°S to 34°S[1]. In addition to the contributions of preformed nutrients

to the total nutrient concentrations of 24 ± 2% in the NBUS and 38 ± 2% in the SBUS, this amounts to a CO₂ uptake by the utilization of $N_{pref}$ of 16.3–58.8 Tg C year⁻¹ for the NBUS, and 5.1–16.0 Tg C year⁻¹ for the SBUS. Hereby, despite of the inherent methodology that could be held responsible for the discrepancy in the estimated new production rates, they provide lower and upper cases to assess the magnitude of the effect preformed nutrient consumption may hold in the BUS. Hence, given these lower (14.5 + 7.5 = 22) and upper (58.8 + 16.0 = 74.8) estimates, a new production driven by the utilization of $N_{pref}$ of ~22–75 Tg C year⁻¹ implies that the biological carbon pump in the BUS countervails 20 up to 68% of the CO₂ release from the biological carbon pump within the Atlantic sector of the Southern Ocean between 44° and 58°S of ~110 Tg C year⁻¹[15,16] (Fig. 5). These results emphasize the role of the BUS as a significant hub for restoring the CO₂ uptake efficiency of the biological carbon pump. However, its strength may be impacted by global change processes, such as the response of upwelling intensities to climatic changes, as well as by human impacts through fishery practices. While the first is difficult to estimate due to e.g. the poor representation of the BUS in numerical models[27] and ongoing changes in the ecosystem structure[61,62], the latter refers e.g., to emissions of aqueous CO₂ caused by disturbances to the seafloor through bottom trawling. In relation to the CO₂ uptake of the biological carbon pump (~22–75 Tg C year⁻¹), a release of sedimentary carbon through bottom trawling of around 5 Tg C year⁻¹ as estimated for the BUS[63] could lead to significant disruptions of the system, with as yet unknown consequences for the biological carbon pump that will need to be addressed in future studies.

## Methods

### Study region

The Benguela Upwelling System (BUS) stretches across the north of Cape Frio from ~15°S to Cape Agulhas (~35°S), while being bound by warm waters of the Angola and Agulhas current to its northern and

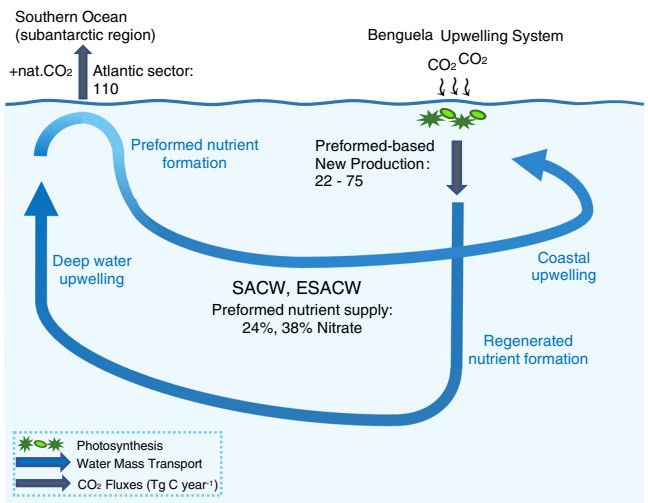

**Fig. 5 | Schematic overview on the coupling between Southern Ocean CO₂ loss and CO₂ uptake in the Benguela Upwelling System through the biological carbon pump.** $CO_2$ fluxes (grey arrows) of the biological carbon pump at the air-sea interface and subsurface, regulated by the effects of preformed-based new production in the Benguela Upwelling System and natural outgassing of $CO_2$ in the Atlantic sector of the Southern Ocean (110 Tg C year⁻¹ [15,16]). The transport of water masses from Southern Ocean across the South Atlantic (blue arrows) shapes the supply of preformed nutrient inventories of the upwelling source water masses (SACW South Atlantic Central Water, ESACW Eastern South Atlantic Central Water) of the northern and southern Benguela Upwelling System. The figure was generated using Inkscape[107].

southern ends, respectively[47]. Coastal upwelling is thereby controlled by south-easterly winds emanating from the interplay between the South Atlantic Anticyclone (SAA) and the continental low pressure trough, causing the emergence of distinct upwelling cells along the shoreline[46]. The strongest upwelling cell is located at Lüderitz (~26°S) and separates the northern (NBUS) from the southern (SBUS) upwelling region[35–37]. One of the main upwelled waters is South Atlantic Central Water (SACW), which represents a Sub-Antarctic Mode Water that itself is a mixture of Antarctic Intermediate Water and Subtropical Mode Water[57–59]. These water masses subduct beneath warmer subtropical surface waters north of the Sub-Antarctic Front around 36°S–54°S and are transported eastward as SACW across the South Atlantic into the Cape basin[64–66]. Here, SACW converges with the Agulhas water from the Indian Ocean to form ESACW that enters the SBUS from the south[67–69]. The majority of the SACW circumvents the SBUS along the Benguela Current and enters the BUS from the north via the Angola-Benguela Frontal Zone (ABFZ) as a poleward undercurrent, which, unlike ESACW, is nutrient-enriched and oxygen-depleted[70].

Upwelling intensities in both subsystems exhibit a seasonal pattern which is mainly driven by temporal shifts of the SAA. As the SAA moves north-westward, the NBUS experiences maximum upwelling intensities during austral winter (June-August), while leading to a dominating westerly wind regime which weakens upwelling in the SBUS. With the south-eastward displacement of the SAA, upwelling in the SBUS is mainly confined to the summer season (September-March)[38,47,71]. Additionally, the seasonality in upwelling is more pronounced in the SBUS as compared to the NBUS due to the perennial upwelling-favourable winds that reign in the northern region[46]. This is also reflected in primary productivity which in the SBUS is twice as large in the summer than winter[72]. Furthermore, studies have shown a decrease and intensification of upwelling in the NBUS and SBUS, respectively, over the past decade in response to global warming[44,52] and a southward shift of the SAA[73]. The associated increase in sea water temperatures in the NBUS[62] could potentially be a cause of the

prominent decrease in zooplankton size spectra in the BUS[61]. Nevertheless, the productivity has remained consistently elevated, and still supports high fishery yields in both subsystems with sardine and horse mackerel as main target species in the SBUS and NBUS, respectively[74–76].

## Sea surface pCO₂ data collection

For the analysis and quantification of air-sea gas exchanges within the BUS, continuous underway measurements were carried out as described in ref. study[31] between 2008 and 2019 (Supplementary Table 1). Hereby, various instruments were utilized to analyse the sea surface partial pressure of $CO_2$ and were fed with seawater by the vessel's circular pumps or autonomous pump systems, drawing water from ~5-7 m from the vessel's moon pool or at its bows. The devices comprise of a LI-7000 $CO_2/H_2O$ analyser (Licor Biosciences), which measured $pCO_2$ by equilibration of surface water with air and the detection of the equilibrium concentration of $CO_2$ in air during RV Maria S. Merian cruise 7/2. A Pro-Oceanus Systems Inc. PSI $CO_2$-Pro™ was applied during RV Meteor cruise 76/2, using gas equilibration and infra-red detection of the $CO_2$ gas stream while being linked to the FerryBox flow-through system. During the remaining cruises, an underway Carbon Dioxide Analyzer (SUNDANS, Marianda) equipped with the infra-red sensor LI-7000 was applied to measure the mole fraction of $CO_2$ in seawater ($xCO_2$), which was converted to $pCO_2$ using underway records of atmospheric pressure, Sea Surface Temperatures and the equilibrator temperature of the SUNDANS system. Additional quality-controlled measurements on sea surface $fCO_2$ from the Surface Ocean $CO_2$ Atlas (SOCAT) v2020[33] were converted into $pCO_2$[77] and embedded into our analysis. All $pCO_2$ measurements were normalized to a reference year (2020)[78] by using a mean yearly change rate in seawater $pCO_2$ of 1.5 μatm year⁻¹ for observations up to year 1992 and 1.9 μatm year⁻¹ for the more recent time period according to the updated oceanic $pCO_2$ trend of Takahashi et al.[79] and multiplying it with respective observations for both upwelling regions (NBUS, SBUS). Overall, the extended data set on $pCO_2$ records used in this study is homogeneously distributed in space as it covers the shelf and coastal areas along the continental margin off Namibia and South Africa, while spanning a timeframe from 1986 to 2020 with over 250 000 data points within the area from approximately 5°E to 18.7°E, and 16°S to 34.5°S. All normalized measurements were spatially interpolated on a 0.1° x 0.1° grid using ordinary kriging as a geostatistical technique, which allows us to perform an error propagation of the gridding procedure and to account for the spatial autocorrelation of $pCO_2$. The uncertainty in average estimates thereby provides an outline on the strong variability of $pCO_2$ that can be found in coastal upwelling settings. In addition, our $pCO_2$ climatology offers an updated view on $CO_2$ sources and sinks in comparison to previous $pCO_2$ climatologies[77,80] which were merely based on the SOCAT dataset that largely misses $pCO_2$ recordings in the NBUS coastal region (Fig. 1a). To reduce the uncertainty in the gridding process, we only applied ordinary kriging to those grid cells with sufficient data coverage. Ordinary kriging was performed using the R automap package[81], which incorporates geostatistical routines from gstat[82]. In a first step, we computed a sample variogram to outline the spatial correlation of observations as a function of distance, and added a model to fit the spatial variation of $pCO_2$ (Supplementary Fig. 3). In a next step, predictions were made for the designated grid cells by including kriging weights that were formed on the basis of the fitted model and $pCO_2$ measurements within the surrounding neighbourhood, while corresponding error estimates were provided as variances ($v$) and standard deviations ($\sigma = \sqrt{v}$) for each grid cell. As the performance of ordinary kriging is time intensive, we only used observations within a maximum of 0.5° spherical distance from the prediction location to speed up the process, without noting any influence on the resulting predictions or error estimates. The total mean uncertainty (SE) of $pCO_2$ for the NBUS and SBUS is

given as the standard error, which results from a combination of measurement error, spatial variance and the gridding procedure. It is derived by dividing the sum squares propagation by the square root of its degrees of freedom (Eq. 1), which represents the number of grid cells ($N$). To account for spatial autocorrelation, we corrected $N$ by replacing it with the effective number of grid cells ($N_{eff}$) following Landschützer et al.[83]

$$SE = \frac{\sqrt{\sum_{i=1}^{N} \sigma_i^2}}{\sqrt{N_{eff}}} \tag{1}$$

We estimated $N_{eff}$ by using autocorrelation length scales for $pCO_2$ that were derived from the variogram as the distance (range) where the spatial dependence (semi-variance) levels off (Supplementary Fig. 3). By calculating the point-to-point distance of each grid cell within both upwelling regions, we could determine the effective number of grid cells outside of the autocorrelation length scale. We estimated short length scales in the range of around 0.01° to 1.4° spherical distances, which, in kilometres, resembles those found in coastal regions (~50 km) as a result of the heterogeneity of the water masses and physical turbulence caused by upwelling[84].

## Carbon flux calculation and its uncertainties

Differences in the partial pressure of carbon between the sea surface ($pCO_{2,sw}$) and atmosphere ($pCO_{2,at}$) were used to determine carbon flux rates ($FCO_2$) using Eq. (2):

$$FCO_2 = K_0 * k * (pCO_{2,sw} - pCO_{2,at}) \tag{2}$$

where $K_0$ is the solubility coefficient of $CO_2$[85] and $k$ represents gas transfer velocity of $CO_2$[86]. The gas transfer velocity $k$ was calculated following Eq. (3):

$$k = 0.251 * u^2 * \left(\frac{Sc}{660}\right)^{-0.5} \tag{3}$$

Sc is the Schmidt number of $CO_2$ in seawater, 660 represents Sc at 20 °C water temperature and $u$ refers to the wind speed (m/s) at 10 m above the sea surface. Additional data on sea surface temperature (SST) (°C) and salinity (PSU) were thereby required for the determination of Sc using the parameterization after Wanninkhof[86]. Data on wind speed, SST and salinity were based on shipboard measurements (Supplementary Table 1), which were spatially interpolated on a 0.1° x 0.1° grid based on the ordinary kriging procedures as outlined in the previous section for $pCO_2$. The flux calculation was then performed using the average sea surface $pCO_2$, wind speed, SST and salinity of the kriging predictions within the defined NBUS and SBUS region. Seasonal flux estimates were calculated using kriged and averaged $pCO_2$, wind speed, SST and salinity data collected during spring and summer (September till March), and austral autumn and winter (April till August), respectively. The total mean uncertainty of the individual parameters was estimated using Eq. 1. In two cases of the NBUS austral summer season, the variograms depict long autocorrelation length scales, resulting in a relatively low effective number of grid cells. For these cases, we used the uncorrected number of grid cells $N$ to calculate the total mean uncertainty (entries marked with * in Supplementary Table 2).

To address another source of uncertainty in the air-sea gas exchange, we further calculated the gas transfer velocity $k$ and subsequent fluxes on the basis of other formulations for $k$ given by Wanninkhof[87] (hereafter KW92), Wanninkhof & McGillis[88], Nightingale et al.[89], McGillis et al.[90], as well as Ho et al.[91] with differing wind speed dependencies (Supplementary Table 3). The resulting flux values based on KW92[87] and Wanninkhof (hereafter KW14)[86] thereby represent the highest and lowest estimates, respectively, and differ on average by ~70% in both subsystems. The parameterization after KW14[86] foresees the use of a wind speed product with high temporal resolution (6-hour), with estimated flux values resembling those derived from previous formulations of $k$, which were based e.g., on dual tracer methods conducted in the North Sea (Nightingale et al.[89]) and Southern Ocean (Ho et al.[91]). The KW92 parameterization is thereby based on an outdated $^{14}C$ inventory for the global ocean, while Wanninkhof & McGillis[88] assumed a cubic instead of the commonly used quadric dependency between gas transfer and wind speed[92].

## Water column sampling and analysis

The analysis of water mass characteristics and biogeochemical settings in the BUS was based on data gathered during the various cruises that were partially embedded into the analysis of carbon fluxes (Supplementary Table 1). In addition, we added data from the Global Ocean Data Analysis Project version 2.2020 (GLODAPv2_2020) and data collected during our most recent cruise with RV Sonne (SO285), which took place from 20th August to 2nd November 2021. The sampling was performed with a CTD/Rosette sampler at stations covering on- and offshore areas of both subsystems (Supplementary Fig. 4), allowing a direct comparison of both upwelling zones. We collected CTD profiles of temperature, salinity and oxygen, and defined the upwelling SACW and ESACW source waters by using the potential temperature ($\theta$) definition provided by ref. 93. (Eqs. 4 and 5):

$$SACW : \theta = 8.56 * Salinity - 289.08 \tag{4}$$

$$ESACW : \theta = 9.44 * Salinity - 319.03 \tag{5}$$

The analysis of dissolved inorganic nutrients (phosphate P, nitrate N) was carried out as outlined in ref. 49., with samples from the CTD/Rosette being filtered through disposable syringe filters (0.45 μm) after sampling, filled in pre-rinsed 50 ml PE bottles that were subsequently measured on-board or kept frozen at −20 °C until being analysed in the shore-based laboratory after the expedition. The measurements were performed with a continuous-flow injection system (Skalar SAN plus System) according to methods outlined by Grasshoff et al.[94]. Furthermore, the calculation of preformed and regenerated nutrients ($P, N_{pref}$ and $P, N_{reg}$ respectively) was performed following ref. 95. by including the apparent oxygen utilization (AOU) as well as the oxidation ratios $R_{P:O_2} = 1/138$ (phosphate) and $R_{N:O_2} = 16/138$ (nitrate), using Eqs. (6) and (7):

$$P, N_{reg} = R_{P,N:O_2} \, AOU \tag{6}$$

$$P, N_{pref} = P, N - P, N_{reg} \tag{7}$$

For comparative reasons, we chose the traditional Redfield oxidation ratio to calculate preformed and regenerated nutrient concentrations. For the analysis of total alkalinity (TA) and dissolved inorganic carbon (DIC), samples were collected in 250 ml borosilicate bottles using silicone tubes (Tygon). The bottles were rinsed twice, filled from the bottom to avoid bubbles and analysed on board using the VINDTA 3 C system (Marianda, Kiel, Germany). For TA analysis, the samples were titrated with a fixed volume of hydrochloric acid (HCl) by equal increments of HCl (0.1 N HCl). The analysis of DIC was performed using the coulometric method (Coulometer CM 5015) after $CO_2$ was extracted out of the water sample. During cruise M153, the analysis of DIC was performed with a cavity ringdown spectrometer (Picarro G2201-I, 1510CFIDS2047_v1.0) attached to a Liaison A0301 and an AutoMate Prep device. Both the Picarro and VINDTA 3 C systems were calibrated using Certified Reference Material provided by A. Dickson

(Scripps Institution of Oceanography, La Jolla, CA, USA) for quality assurance.

## (Non-) thermal component analysis of sea surface pCO₂

We reconstructed the non-thermally controlled $pCO_2$ based on our $pCO_2$ climatology, which includes annual and seasonal gridded fields on $pCO_2$, SST and SSS as derived from ordinary kriging interpolations (see previous sections). We thereby applied the $CO_2SYS$[96,97] programme, using sea surface TA and DIC concentrations as input parameters. Due to a lack of shipboard underway measurements for TA and DIC that are needed to resolve the carbonate system, we first reconstructed sea surface TA by leaning on the TA-salinity relationship, which refers to the control of surface TA by freshwater addition or removal that can be mirrored through changes in salinity[98,99]. We therefore performed a linear regression analysis based on TA and salinity gathered from underway shipboard records during cruise SO285, and CTD surface profiles from all cruises listed in Supplementary Table 4. Based on the slope of the linear regression line (Supplementary Fig. 5), we then determined the corresponding TA for each grid cell that contained spatio-temporal interpolated salinity values. Next, we reconstructed sea surface DIC with $CO_2SYS$[96,97] for each grid cell that contained values for TA and the corresponding $pCO_2$, sea surface temperature and salinity. For the performance of $CO_2SYS$[96,97] calculations, we used the dissociation constants of Mehrbach et al.[100] as refit by Dickson and Millero[101] and the borate dissociation constant of Dickson[102]. In a final step, we recalculated sea surface $pCO_2$ by applying $CO_2SYS$[96,97] with all available grid cell values of TA and DIC within the NBUS and SBUS for each season (annual, spring-summer, autumn-winter). By using the respective temperatures and salinities from the average source water mass characteristics (Supplementary Table 4) instead of those from the sea surface, we excluded the warming of upwelled waters during their offshore flow. The surface warming effect on the air-sea gas exchange can ultimately be elaborated by comparing the recalculated, so-called non-thermal $pCO_2$ with the one derived from the actual shipboard measurements. Hereby it should be noted that the number of grid cells between the non-thermal and measured $pCO_2$ can differ due to the difference in the availability of $pCO_2$ measurements and salinity records (see $n$ in Supplementary Table 2) that were used for $CO_2SYS$[96,97] calculations. In case of the SBUS autumn-winter season, this resulted in a relatively high standard deviation of the non-thermal $pCO_2$ compared to the measured $pCO_2$, as there were fewer values available to determine the latitudinal averages (Fig. 4c).

## Statistical information

The uncertainty in average parameter calculations of $pCO_2$, $CO_2$ exchange coefficients and fluxes (Supplementary Table 2), as well as in biogeochemical characteristics of source waters (Supplementary Table 4) is presented as the standard error (s.e.), together with the number of values ($n$) used for the average calculation.

## Data availability

The underlying data on sea surface $pCO_2$ and water column characteristics used in this study are available in the PANGAEA database under the accession code as listed in Supplementary Table 5. Furthermore, our study includes $pCO_2$ records obtained from the Surface Ocean $CO_2$ Atlas (SOCAT) v2020 accessible under https://www.socat.info/index.php/data-access/. Average atmospheric concentrations of $CO_2$ were obtained from the Global Monitoring Laboratory (GML) of the U.S. National Oceanographic and Atmospheric Administration (NOAA) Research (https://www.esrl.noaa.gov/gmd/ccgg/trends/data.html). GLODAP version 2.2020 are freely accessible under the National

Centers for Environmental Information (https://www.ncei.noaa.gov/access/ocean-carbon-data-system/oceans/GLODAPv2_2020/). In addition, the underlying data to reproduce each figure and table are available in Figshare under the accession code https://doi.org/10.6084/m9.figshare.21436494.

## Code availability

The code used to generate the main output of this study is available in Figshare under doi:10.6084/m9.figshare.21436494. The $CO_2SYS$ program for Python as applied in this study is freely available and documented under https://pypi.org/project/PyCO2SYS/.

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

## Acknowledgements

We would like to thank all scientists, technicians, captains and crew members for their support during the various cruises that were embedded in this work. F. Hüge and M. Birkicht are acknowledged for their support in the laboratories. In addition, we would like to thank Peter Landschützer for the discussions on $pCO_2$ mapping strategies and his guidance through kriging procedures. This research was funded by the German Federal Ministry of Education and Research (BMBF) under the grant no. 03F0797A. The Surface Ocean $CO_2$ Atlas (SOCAT) is an international effort, endorsed by the International Ocean Carbon Coordination Project (IOCCP), the Surface Ocean Lower Atmosphere Study (SOLAS) and the Integrated Marine Biosphere Research (IMBeR) program, to deliver a uniformly quality-controlled surface ocean $CO_2$ database. The many researchers and funding agencies responsible for the collection of data and quality control are thanked for their contribution to SOCAT. We also thank P. Wessels and W.H.F. Smith for the provision of the Generic Mapping Tools (GMT). All calculations were executed with R (v.3.5.1)[103] and the Python software (v.3.7)[104].

## Author contributions

T.R. and N.L. designed the study. The shipboard data collection and analysis in 2008 to 2014 was performed by T.R., and in 2019 and 2021 by T.R. and C.S.. C.S., T.R., N.L., A.K.V.d.P., D.C.L., T.L. and K.P. discussed the results and contributed to the writing of the manuscript, while C.S. led the writing of the manuscript.

## Funding

## Competing interests

The authors declare no competing interests.
