## [Peer Review File · Nature Communications]

Regional and global impact of CO₂ uptake in the Benguela Upwelling System through preformed nutrientsReviewer #1 (Remarks to the Author):

Review of manuscript "Biological carbon pump affected by CO₂ uptake in the Benguela Upwelling system" by Siddiqui, Rixen, Lahajnar, Van der Plas, Louw, Lamont, and Pillay

The authors use shipboard measurements complemented with data from an updated version of the SOCAT database (v2020) to show that the Benguela Upwelling System (BUS) acts as a CO₂ source in its northern part and a CO₂ sink in the southern region. Overall, the manuscript is well organized and easy to follow. It includes a nice description of the oceanography in the BUS and adjacent regions that provides context for understanding the results that are presented. The use of a consistent metric for the upwelling zone's boundary makes the characterization of the conditions in the BUS more robust and facilitates comparisons between regions (and studies). The analysis to quantify the effect of different factors (biological uptake, warming, preformed and regenerated nutrients) on pCO₂ is compelling and the results support the manuscript's main conclusion that the intensification of the biological carbon pump due to higher supply of preformed nutrients from the Southern Ocean into the southern part of the BUS offsets the increase in pCO₂ caused by surface warming, turning that region into a net sink of CO₂. The study also quantifies the CO₂ sink due to new production and estimates that new production in the BUS can offset approximately 20% of the CO₂ outgassing in the Atlantic sector of the Southern Ocean, making the BUS a globally significant CO₂ sink.

However, the writing is bit vague or confusing in some parts and the text contains a few inconsistencies. The specific comments below are suggestions to improve clarity:

Line 44: The word "loss" is vague. I would change it to "outgassing".

Line 59: This sentence is awkward. I would change "turned e.g." to "makes".

Line 65: If I understand this sentence correctly, the value for "CO₂ invasion" is the difference between 740 Tg C/year and 400 Tg C/year, so it should be 340 Tg C/year (not 380).

Line 70: I would change "restore" to "increase".

Line 77: This sentence is a bit awkward. I would change "hardly improved these conditions" to "have not resolved these issues".

Line 80: I would change "model outcomes" to "model simulations".

Lines 125–126: I would change "according to the opposing signs" to "given the opposite signs of the flux in the two regions".

Lines 143–144: This sentence is awkward and confusing. Do the authors mean to say that nitrogen fixation is limited by low phosphate concentrations? This sentence should be rewritten to clarify its meaning.

Line 195: If I understand this sentence correctly, the 20% figure comes from dividing the low end of CO₂ uptake by new production (19.7) by the CO₂ release in the Southern Ocean (110). However 19.7 / 110 is approximately 18% and not "over 20%". This sentence should be rewritten to be more precise.

In summary, I find that the manuscript is acceptable for publication after minor revisions.

Reviewer #2 (Remarks to the Author):

This manuscript presents a compilation of shipboard observations from the nearshore and offshore portions of the Benguela Upwelling System (BUS) in the eastern South Atlantic Ocean, which is a societally-important, highly-productive eastern boundary upwelling system that has traditionally

been undersampled. Using these data, the authors then draw several conclusions regarding 1) the mean and seasonal cycle of surface ocean pCO₂ and air-sea CO₂ flux in the Northern and Southern portions of the BUS; 2) the mechanisms that control the observed pCO₂ values; 3) the role of the BUS in the global carbon cycle. The new data fill an important gap in existing observations, particularly in the northern BUS shelf region, and provide an updated view of carbon fluxes in this highly variable, dynamic region. Unfortunately though, the analysis of the data suffers from a number of severe shortcomings, which raises doubts about the validity of the conclusions of the study. With fundamental changes to the analysis methods and a complete reworking of the text, however, this study has potential to make an important contribution to our understanding of the role of eastern boundary upwelling systems in the oceanic carbon cycle.

-Of the concerns I have with the data analysis methods used in this study, one of the most serious is the failure to properly account for the spatio-temporal variability in the sampling and in the underlying fields. This could have a significant impact on the conclusions reached here. For instance, the use of the standard error formula to determine the uncertainty in the quantities in Tables 1 and 2 is problematic when $n = \text{number of grid points}$ (as is done here), because for these quantities that are highly correlated in space and time, the degrees of freedom does not equal the number of grid points. In another example, the pCO₂ dataset used in this study is clearly biased towards spring and summer, in terms of the number of observations (Figure S1); yet the authors apparently just average all the data together, regardless of this seasonal sampling bias and seasonal differences in the underlying fields, to determine the annual average. Similarly, the results shown in Figure 2c give the mean and standard deviation as a function of latitude, but clearly there is considerable noise given the choice of bin size. In all of the plots shown in Figure 2, one wonders if the data are normally distributed; perhaps a two-dimensional histogram would be more illuminating in this regard.

-Related to this issue, the discussion of the data and the driving mechanisms should be more careful in the treatment of variability at other spatial or temporal scales, which could be influencing what is observed. For example, there is no consideration of interannual variability in either the pCO₂ calculations or the water mass characteristic analysis. How representative are the nutrient values determined from a single cruise?

-The authors seem to compute CO₂ fluxes using Equations (1) and (2) with the wind speed, SST, and SSS obtained from cruise measurements that have then been box-averaged on a 0.1-degree grid. However, accounting for variability in wind speed in particular is very important for accurately estimating gas fluxes. Equation (2) was developed by Wanninkhof 2014 for 6-hourly wind speeds and is very likely inappropriate to use with 6 month averaged wind speeds (see discussion in Wanninkhof 2014). Additionally my concerns regarding how representative the cruise data are and how appropriate box-averaging is for pCO₂ applies even more strongly to wind speed. This seems to be a serious issue with the methodology of this aspect of the study.

-The source water for the BUS (SACW and ESACW) is subtropical mode water (lines 217-220), yet the motivation of the study and the discussion of the results rely heavily on a link to the preformed nutrients created in the Southern Ocean, which are, according to line 68, "transported northward and subducted beneath warmer and lighter subtropical water masses". This manuscript completely neglects any steps between the preformed nutrients being created in the Southern Ocean in Subantarctic Mode Water and the nutrients in the Subtropical Mode Water reaching the BUS, yet these potentially very important and transformative processes are not well-known. To me, this represents a significant missing link in the explanation presented here.

-One of the main quantitative results is that the BUS accounts for new production driven by preformed nutrient utilization of 19.7-71.1 Tg C year⁻¹ (line 189), which is a very large range. It would be helpful to have a better discussion of what is causing that spread. It is also unclear how that result relates to the analysis of the effect of consumption of preformed nutrients on pCO₂ in the previous paragraph. There, the authors argue that consumption of preformed nutrients has a slightly bigger impact in the SBUS than in the NBUS, yet the new production driven by performed nutrients (estimated in the paragraph starting on line 184) is almost entirely attributable to the NBUS. How do the authors reconcile this?

-The authors motivate their study primarily with a discussion of the role of the global-scale biological pump and preformed nutrients in determining atmospheric CO₂ concentrations. I find this part of the paper not very clear and only tenuously related to the actual results presented here. My suggestion would be to shift the focus more towards the BUS itself and less on the Southern Ocean and the global carbon cycle. In addition, the "biological pump" and "preformed nutrients" should be defined more explicitly when they are first used.

-Overall the writing of this paper needs to be improved substantially. Currently there are numerous spelling and grammar errors and many instances of imprecise and/or inaccurate language use.

-The organization of the manuscript also should be overhauled. Some of the information in the Study Region part of the Methods, especially regarding the definition of the NBUS and SBUS, would be very helpful at the beginning of the paper, because most readers will not be familiar with this area. Similarly, most of the information in Tables 1 and 2 would fit much better in the supplementary information, while Supplementary Figures 2 and 3 would be much more impactful in the main text. The end of the main text is very abrupt and inconclusive.

Minor comments:

-The title of the manuscript suggests that CO₂ uptake in the BUS impacts the biological carbon pump, while the causal relationship is actually reversed -- the biological carbon pump affects CO₂ uptake.

-The use of the name "Antarctic Divergence" on line 61 is unnecessary and should be removed for a more general audience.

-How sensitive are the results to the choices made in defining the upwelling zone boundary (i.e., the values on line 98)? What about the choice of 117:1 as the stoichiometric ratio for C:P (line 161)?

-Why is there an "approximate" sign in front of the factor of 0.12 used on line 164. What was approximate about the number that was used?

-Given that the cruise RV Meteor M153 is the only data source for the water mass analysis presented here, more details should be given about this cruise in the text / on Figure 1. Where/when were these data collected from? How representative are these values, compared to e.g., GLODAP, in this region?

-Southern Ocean carbon uptake estimates given on lines 64-66 are just one such set of estimates, and there is a lot of variability. Why not use a more recent estimate, such as those from Landschutzer et al.? At the least, some discussion is needed about the uncertainty in Southern Ocean CO₂ flux estimates.

-The interpolated map shown in Figure 1c is not used in the discussion and in my opinion adds nothing to the paper that is not already shown in Figure 1b.

-Line 268: the definition used in ref 71 to determine the SACW and ESACW should be explicitly given.

-This may be just a matter of taste, but I disagree with the terminology of "leakage" when discussing outgassing of natural CO₂ from the deep ocean that is associated with incomplete nutrient utilization. Calling it a leakage implies an accidental or undesired nature, whereas the biologically-mediated transfer of carbon to the deep ocean and the outgassing of remineralized carbon are both necessarily in balance on a global scale, if the system is in steady state.

-The comparison to a previously published study for carbon uptake in the NBUS and SBUS on line 130 is helpful, but it would be best to mention what type of study that was (i.e., observation or model-based) and any relevant differences.

-The choices of various dissociation constants to use in the carbonate system calculations (in CO2SYS) should be stated.

Review by Alison Gray, 28-May-2021

Reviewer #3 (Remarks to the Author):

This paper addresses the role of the Benguela upwelling system (BUS) in regulating the ocean uptake of CO₂. The authors use a new compilation of shipboard measurements over the past two decades that significantly increases the spatial/temporal coverage offered by SOCAT for pCO₂, particularly in the coastal areas. The authors claim the BUS act as a source of CO₂ in its northern portion and as a sink of CO₂ in its southern counterpart. The difference between the two sectors is due to a higher portion of preformed nutrients in the upwelling water in the south with respect to the north. This results in the biological carbon pump being more effective at decreasing surface pCO₂.

The objective of the paper are relevant for the wide climate science community and the paper has the potential to be a significant contribution to the field. However, despite the design of the analysis seems sound, its execution and description are very confusing with important contradictions in the parts illustrating the main reasoning supporting the conclusions.

My main concern is about lines 184-202 – This paragraph is very confusing and because it contains the main reasoning supporting one of the highlights of the paper, it needs some improvement. First of all, it uses some estimate of the volume of upwelling water which is not reported anywhere on the paper. It should be explicitly reported in one of the tables, for example. Moreover, it starts talking about estimating “the potential amount of CO₂ transfer into the ocean interior on the basis of new production rates from the BUS...” but it gives an estimate of total primary production first (180-650 g C/m²/year) for the NBUS which integrated over the area amounts to 68-245 Tg C/year. Then it gives the actual new production estimates based on the P_{pref} share in NBUS source water reported in Table 2 (29%) , which results in 19.7-71.1 Tg C/year.

Now, it moves to SBUS and it gives directly the estimate of new production without going through the calculations like it did for NBUS. This estimate is 2 orders of magnitude lower than for NBUS. This is weird because nutrient concentration, preformed content and area do not justify such reduction with respect to NBUS, so I assume it must be the volume of upwelling water. Therefore, this should be explicitly reported in one of the tables. It also states “The contribution of P_{pref} to new production from SBUS is low” but the content of P_{pref} reported in Table 2 for this region is around 47% (i.e. way higher than for NBUS).

Then it goes on suggesting that the new production in the NBUS (19.7 – 71.1 Tg C/year) is the one that countervails over 20% of the CO₂ loss in the subpolar South Atlantic region. This means that the compensation over the South Atlantic CO₂ loss happens through the NBUS rather than the SBUS, which is the opposite of what is stated throughout the paper.

So, overall this reasoning appears confusing and it should be better explained before considering this paper for publication.

Further comments:

Lines 42-43. Not clear what “their formation refers to”. Is it the formation of source waters in the Southern Ocean? Or the formation of upwelling source waters? The formation of preformed nutrients? Consider re-phrasing.

Lines 64-69. The Atlantic sector is said to be responsible of 34% of the natural CO₂ release (~110 Tg C/year). I understand this is 34% of the total natural CO₂ release from the Southern Ocean, which is reported here to be about 400 Tg C/year. So, isn't it just about 27.5% ?

In Figure 1 it would be useful to see the limits defined between SBUS and NBUS.

Lines 134 – Perhaps give this number in Tg C /year for comparison.

Lines 168-169 – The calculated pCO₂ for NBUS is 50 uatm lower than the one estimated from field measurements. This is non-negligible (11.4%) as it would make the NBUS a weaker source of CO₂. It would be interesting if the authors could explore what are the possible reasons for this discrepancy.

The choice of figures to include in the main text is not very representative of the main results. I would consider substituting one of the two figures with one currently in supplementary that would better guide the reader through the reasoning of the paper (either S2 or S3).

Response letter to the referees' comments

We hereby respond to the various comments and concerns raised during the peer review process of our former manuscript entitled "Biological carbon pump affected by CO₂ uptake in the Benguela Upwelling System", which we revised as outlined below.

Reviewer #1

The authors use shipboard measurements complemented with data from an updated version of the SOCAT database (v2020) to show that the Benguela Upwelling System (BUS) acts as a CO₂ source in its northern part and a CO₂ sink in the southern region. Overall, the manuscript is well organized and easy to follow. It includes a nice description of the oceanography in the BUS and adjacent regions that provides context for understanding the results that are presented. The use of a consistent metric for the upwelling zone's boundary makes the characterization of the conditions in the BUS more robust and facilitates comparisons between regions (and studies). The analysis to quantify the effect of different factors (biological uptake, warming, preformed and regenerated nutrients) on pCO₂ is compelling and the results support the manuscript's main conclusion that the intensification of the biological carbon pump due to higher supply of preformed nutrients from the Southern Ocean into the southern part of the BUS offsets the increase in pCO₂ caused by surface warming, turning that region into a net sink of CO₂. The study also quantifies the CO₂ sink due to new production and estimates that new production in the BUS can offset approximately 20% of the CO₂ outgassing in the Atlantic sector of the Southern Ocean, making the BUS a globally significant CO₂ sink.

We thank the reviewer for valuing our work and for stating the study's main conclusions to be comprehensive and compelling.

However, the writing is bit vague or confusing in some parts and the text contains a few inconsistencies. The specific comments below are suggestions to improve clarity:

Line 44: The word "loss" is vague. I would change it to "outgassing".

We made the according changes as suggested.

Line 59: This sentence is awkward. I would change "turned e.g." to "makes".

We made the according changes as suggested.

Line 65: If I understand this sentence correctly, the value for "CO₂ invasion" is the difference between 740 Tg C/year and 400 Tg C/year, so it should be 340 Tg C/year (not 380).

We apologize for this mistake and made the according changes as suggested.

Line 70: I would change "restore" to "increase".

Line 77: This sentence is a bit awkward. I would change "hardly improved these conditions" to "have not resolved these issues".

We made the according changes as suggested.

Line 80: I would change "model outcomes" to "model simulations".

We made the according changes as suggested.

Lines 125–126: I would change "according to the opposing signs" to "given the opposite signs of the flux in the two regions".

We made the according changes as suggested.

Lines 143–144: This sentence is awkward and confusing. Do the authors mean to say that nitrogen
fixation is limited by low phosphate concentrations? This sentence should be rewritten to clarify its
meaning.

We restructured the paragraph and dismissed this sentence, as we wanted to highlight the absence of
preformed Nitrate and its implication for the CO₂ source behaviour of the Peruvian Upwelling System.

Line 195: If I understand this sentence correctly, the 20% figure comes from dividing the low end of
CO₂ uptake by new production (19.7) by the CO₂ release in the Southern Ocean (110). However 19.7 /
110 is approximately 18% and not “over 20%”. This sentence should be rewritten to be more precise.

We restructured the paragraph and gave the percentage of our revised calculations as suggested.

In summary, I find that the manuscript is acceptable for publication after minor revisions.

**Reviewer #2**

This manuscript presents a compilation of shipboard observations from the nearshore and offshore
portions of the Benguela Upwelling System (BUS) in the eastern South Atlantic Ocean, which is a
societally-important, highly-productive eastern boundary upwelling system that has traditionally been
undersampled. Using these data, the authors then draw several conclusions regarding 1) the mean and
seasonal cycle of surface ocean pCO₂ and air-sea CO₂ flux in the Northern and Southern portions of
the BUS; 2) the mechanisms that control the observed pCO₂ values; 3) the role of the BUS in the
global carbon cycle. The new data fill an important gap in existing observations, particularly in the
northern BUS shelf region, and provide an updated view of carbon fluxes in this highly variable,
dynamic region. Unfortunately though, the analysis of the data suffers from a number of severe
shortcomings, which raises doubts about the validity of the conclusions of the study. With fundamental
changes to the analysis methods and a complete reworking of the text, however, this study has
potential to make an important contribution to our understanding of the role of eastern boundary
upwelling systems in the oceanic carbon cycle.

We thank the reviewer for acknowledging the novelty of this study and its potential to be of valuable
contribution to the understanding of EBUS in the global carbon cycle. We revised the methods by
providing extra analysis on the source water mass characteristics, the flux calculations as well as on
the role of the BUS in the global carbon cycle to overcome the technical concerns and to strengthen the
outcome of our study. More information about the specific changes we made is given below.

-Of the concerns I have with the data analysis methods used in this study, one of the most serious is
the failure to properly account for the spatio-temporal variability in the sampling and in the underlying
fields. This could have a significant impact on the conclusions reached here. For instance, the use of
the standard error formula to determine the uncertainty in the quantities in Tables 1 and 2 is
problematic when n = number of grid points (as is done here), because for these quantities that are
highly correlated in space and time, the degrees of freedom does not equal the number of grid points.
Instead of gridding our data and calculating the arithmetic mean of each grid cell as done in the
previous version of our manuscript, we now applied the ordinary kriging method to account for the
variability and autocorrelation, as well as the degrees of freedom, within our dataset. Ordinary kriging
further allows us to perform an error propagation, which helps us to improve our uncertainty estimation.
More information on the different procedures of the ordinary kriging as applied in this study is given in
the Method section.

In another example, the pCO₂ dataset used in this study is clearly biased towards spring and summer,
in terms of the number of observations (Figure S1); yet the authors apparently just average all the data
together, regardless of this seasonal sampling bias and seasonal differences in the underlying fields, to

determine the annual average. Similarly, the results shown in Figure 2c give the mean and standard
deviation as a function of latitude, but clearly there is considerable noise given the choice of bin size. In
all of the plots shown in Figure 2, one wonders if the data are normally distributed; perhaps a two-
dimensional histogram would be more illuminating in this regard.

Although our pCO₂ dataset covers the coastal and offshore regions of the BUS, we acknowledge the
seasonal sampling bias to be a drawback of our study that we unfortunately could not overcome, and
admit that our annual pCO₂ analysis might be more representative for the austral spring and summer
season. In our study, instead of focussing on the seasonal comparison within a subsystem, we are
more concerned with the difference in pCO₂ between the NBUS and SBUS. Considering the difference
in the size of the areas (NBUS: 377 400 km², SBUS: 177 600 km²) and the rather low difference in grid
points between the NBUS and SBUS, we are convinced that such a comparison of the seasonal pCO₂
between the subsystems is still plausible with our available dataset. Furthermore, we added a
histogram with seasonal resolution for each subsystem in Fig. 2d,e to provide more background
information on pCO₂. The data is partially deviating from a normal distribution as being positively
skewed due to the effect of upwelling events in both subsystems during both seasons. Hence, although
there are more measurements available for the austral spring and summer season, our data mirrors the
upwelling-related variability in pCO₂ for both seasons, and is therefore applicable for analyzing
seasonal pCO₂ dynamics in the BUS. The noise in Figure 2c thereby reflects the variability of pCO₂ that
is created in close proximity to the coast where the impact of upwelling is most prevalent. The choice of
a lower bin size (resolution) would only smoothen our pCO₂ and remove important information on the
spatial variation that is helpful to understand the impact of upwelling as well as the choice of the
upwelling system's offshore boundaries.

-Related to this issue, the discussion of the data and the driving mechanisms should be more careful in
the treatment of variability at other spatial or temporal scales, which could be influencing what is
observed. For example, there is no consideration of interannual variability in either the pCO₂
calculations or the water mass characteristic analysis. How representative are the nutrient values
determined from a single cruise?

In our revised manuscript, we estimated the driving mechanisms of pCO₂ by considering data from
previous cruises to provide a broader insight into the individual source water characteristics and their
variability. A map of the sampling stations during the respective cruises is thereby given in
Supplementary Fig.4. Furthermore, we considered the uncertainties in the source water parameters to
derive upper and lower case scenarios of the pCO₂ simulations to elucidate the sensitivity of our
results.

-The authors seem to compute CO₂ fluxes using Equations (1) and (2) with the wind speed, SST, and
SSS obtained from cruise measurements that have then been box-averaged on a 0.1-degree grid.
However, accounting for variability in wind speed in particular is very important for accurately estimating
gas fluxes. Equation (2) was developed by Wanninkhof 2014 for 6-hourly wind speeds and is very likely
inappropriate to use with 6 month averaged wind speeds (see discussion in Wanninkhof 2014).
Additionally my concerns regarding how representative the cruise data are and how appropriate box-
averaging is for pCO₂ applies even more strongly to wind speed. This seems to be a serious issue with
the methodology of this aspect of the study.

To overcome the technical concerns raised, we additionally calculated the CO₂ fluxes using other
formulars for the piston velocity (k) with differing wind speed dependencies to shed light on our flux
uncertainty, with the results given in Supplementary Table 3. The flux values based on Wanninkhof
(1992)¹ and Wanninkhof (2014)² thereby represent the highest and lowest estimates, respectively, and
differ on average by ~70% in both subsystems. Although the parameterization after Wanninkhof (2014)
foresees the use of a wind speed product with high temporal resolution (6-hour), the estimated flux

values resemble those derived from previous formulations of k , which were based e.g., on dual tracer
methods conducted in the North Sea (Nightingale *et al.*³) and Southern Ocean (Ho *et al.*⁴). Wanninkhof
(1992) is thereby based on an outdated ¹⁴C inventory for the global ocean, while Wanninkhof & McGillis
(1999)⁵ assumed a cubic instead of the commonly used quadric dependency between gas transfer and
wind speed.

-The source water for the BUS (SACW and ESACW) is subtropical mode water (lines 217-220), yet the
motivation of the study and the discussion of the results rely heavily on a link to the preformed nutrients
created in the Southern Ocean, which are, according to line 68, "transported northward and subducted
beneath warmer and lighter subtropical water masses". This manuscript completely neglects any steps
between the preformed nutrients being created in the Southern Ocean in Subantarctic Mode Water and
the nutrients in the Subtropical Mode Water reaching the BUS, yet these potentially very important and
transformative processes are not well-known. To me, this represents a significant missing link in the
explanation presented here.

To address the missing link in our explanation, we highlighted the origin of South Atlantic Central Water
(SACW), which is the water mass that wells up in the NBUS and, after converging with Indian Central
Water, in the SBUS. SACW represents a Sub-Antarctic Mode Water that itself is a mixture of Antarctic
Intermediate Water and Subtropical Mode Water (see *Ref.*⁶⁻⁸). These water masses are then subducted
beneath warmer subtropical surface waters north of the Sub-Antarctic Front around 36°S -54°S and are
transported eastward as SACW across the South Atlantic into the Cape basin. To our understanding,
SACW can therefore be regarded as a water mass that has its origin in the Southern Ocean. The
processes shaping the preformed nutrient concentration of the source water masses finally reaching
the BUS are, as also stated by the reviewer, not well known, but could eventually be addressed using
e.g., Argo float data records from across the South Atlantic basin. In view of our study, although such
an analysis would be beneficial and of valuable contribution, it is beyond the scope of our paper, and
should hence be treated in a separate study.

-One of the main quantitative results is that the BUS accounts for new production driven by preformed
nutrient utilization of 19.7-71.1 Tg C year⁻¹ (line 189), which is a very large range. It would be helpful to
have a better discussion of what is causing that spread. It is also unclear how that result relates to the
analysis of the effect of consumption of preformed nutrients on pCO₂ in the previous paragraph. There,
the authors argue that consumption of preformed nutrients has a slightly bigger impact in the SBUS
than in the NBUS, yet the new production driven by performed nutrients (estimated in the paragraph
starting on line 184) is almost entirely attributable to the NBUS. How do the authors reconcile this?

To provide a better link between our previous paragraph on the effect of preformed nutrients on pCO₂
and the contribution of preformed nutrients to new production, we referred to the different roles that
biological nutrient consumption plays in the marine carbon cycle. Hereby, we note the role of preformed
nutrients in compensating for CO₂ outgassing during their formation at higher latitudes, before going on
with their contribution to CO₂ transfer into the ocean through preformed-based new production.

In our revised manuscript, we multiplied the volume of upwelling waters (NBUS: 0.9 Sverdrup, SBUS:
0.4 Sverdrup) with corresponding nutrient inventories of the source water masses that are given in
Supplementary Table 4 to estimate new production driven by preformed nutrients. Instead of using
preformed Phosphate, we thereby decided to perform the calculations using preformed Nitrate, as it is
more likely to be the limiting factor for biological production in the BUS⁹. After conversion to Carbon
using the Redfield ratio (106:16), the resulting estimates based on preformed nitrate amount to 14.5 Tg
C year⁻¹ for the NBUS, and 7.5 Tg C year⁻¹ for the SBUS. These values imply a higher preformed
nutrient contribution in the NBUS due to the difference in the volume of upwelling waters between the
NBUS and SBUS.

We further estimated the preformed-based new production by using previously published new
production rates and the contribution of preformed nitrate (N_{pref}) as derived from our water mass
characteristics (Supplementary Table 4) to give a more robust estimate on the role of the BUS in the
carbon cycle. Hereby, the published new production rates fall within a comparatively wide range of 68 –
245 T C year⁻¹ for the NBUS and 13.5 – 42 Tg C year⁻¹ for the SBUS¹⁰⁻¹², which in sum covers
published new production rates of 241 Tg C year⁻¹ derived for the entire BUS from 16°S to 34°S¹³.
Given the contribution of preformed nutrients of 24% in the NBUS and 38% in the SBUS amounts to a
CO₂ uptake by N_{pref} utilization of 16.3 – 58.8 Tg C year⁻¹ for the NBUS, and 5.13 – 16.0 Tg C year⁻¹ for
the SBUS.

In the end, given the lower (14.5 + 7.5) and upper (58.8 + 16.0) estimates, we calculated a preformed-
based new production of ~22 – 75 Tg C year⁻¹. This implies that the biological carbon pump in the BUS
has the potential to countervail 20 up to 68% of the CO₂ release from the biological carbon pump within
the Atlantic sector of the Southern Ocean between 44° and 58°S (Fig. 4). Even if one ignores the high
estimates of the preformed-based new production while also taking into account the variability in
Southern Ocean carbon flux rates, our results emphasize the role of the BUS as a significant hub for
restoring the CO₂ uptake efficiency of the biological carbon pump.

-The authors motivate their study primarily with a discussion of the role of the global-scale biological
pump and preformed nutrients in determining atmospheric CO₂ concentrations. I find this part of the
paper not very clear and only tenuously related to the actual results presented here. My suggestion
would be to shift the focus more towards the BUS itself and less on the Southern Ocean and the global
carbon cycle. In addition, the “biological pump” and “preformed nutrients” should be defined more
explicitly when they are first used.

Given the results of our study, we are convinced that an understanding of the factors governing CO₂
emission scenarios in the BUS can only be achieved when the effect of preformed nutrients is being
considered, as they can foster the biologically-mediated drawdown of pCO₂ below atmospheric levels.
The occurrence of preformed nutrients in the upwelling waters is thereby dependent on processes
shaping the biological pump efficiency in the Southern Ocean, meaning, in order to make predictions
about the future of the BUS as an atmospheric CO₂ sink/source, one has to take into account any
changes that have an impact on the nutrient utilization in the Southern Ocean. In addition, the utilization
of preformed nutrients in the BUS for new production can partially compensate for the missed
opportunity in CO₂ sequestration in regions where the biological pump is less efficient. To our
understanding, these important links have been missing so far for the BUS, while more information is
needed on the processes shaping the preformed nutrient supply into the BUS and other Eastern
Boundary Upwelling Systems.

-Overall the writing of this paper needs to be improved substantially. Currently there are numerous
spelling and grammar errors and many instances of imprecise and/or inaccurate language use.

We took into account the other reviewer’s comments on language and corrected the spelling and
grammar errors, and rephrased certain sentences for clarity.

-The organization of the manuscript also should be overhauled. Some of the information in the Study
Region part of the Methods, especially regarding the definition of the NBUS and SBUS, would be very
helpful at the beginning of the paper, because most readers will not be familiar with this area. Similarly,
most of the information in Tables 1 and 2 would fit much better in the supplementary information, while
Supplementary Figures 2 and 3 would be much more impactful in the main text. The end of the main
text is very abrupt and inconclusive.

To clarify the NBUS and SBUS definition, we added the subsystem boundaries into Fig.1c.

Furthermore, we changed the appearance of Figures in the manuscript and included the former
Supplementary Figure 2 and 3 to the main text, while shifting the former Tables 1 and 2 to the
supplementary section.

Minor comments:

-The title of the manuscript suggests that CO₂ uptake in the BUS impacts the biological carbon pump,
while the causal relationship is actually reversed -- the biological carbon pump affects CO₂ uptake.
We considered renaming our manuscript into “Regional and global impact of CO₂ uptake in the
Benguela Upwelling System through preformed nutrients”.

-The use of the name “Antarctic Divergence” on line 61 is unnecessary and should be removed for a
more general audience.

We made the according changes as suggested and removed the term “Antarctic Divergence”.

-How sensitive are the results to the choices made in defining the upwelling zone boundary (i.e., the
values on line 98)? What about the choice of 117:1 as the stoichiometric ratio for C:P (line 161)?
Since we include the uncertainties of our pCO₂ estimates for deriving CO₂ fluxes, as well as the
uncertainties in the source water mass characteristics that were used to analyse driving mechanisms in
our pCO₂ variability, we were able to provide some information on the sensitivity of our results. As for
the CO₂ fluxes, our results imply, despite the uncertainties, an opposing behaviour between the NBUS
and SBUS as a CO₂ source and sink, respectively. As for the choice of the carbon to nutrient ratio, we
adopted the Redfield ratio of 106:16 that was estimated in previous work within the BUS from Flohr *et*
*al.*⁹.

-Why is there an “approximate” sign in front of the factor of 0.12 used on line 164. What was
approximate about the number that was used?

-Given that the cruise RV Meteor M153 is the only data source for the water mass analysis presented
here, more details should be given about this cruise in the text / on Figure 1. Where/when were these
data collected from? How representative are these values, compared to e.g., GLODAP, in this region?
We included a graphic (Supplementary Fig. 4) showing the source water mass sampling locations as
well as a table with average source water mass characteristics during the individual cruises, including
data from GLODAPv2. According to our extended source water mass analysis as reported in
Supplementary Table 4, our cruise data corresponds well with average values from the GLODAP
database, although we noticed a comparatively high variability in DIC concentrations. As for the share
in preformed nutrients, individual cruises resemble the dominance of preformed nutrients in the SBUS
over the NBUS.

-Southern Ocean carbon uptake estimates given on lines 64-66 are just one such set of estimates, and
there is a lot of variability. Why not use a more recent estimate, such as those from Landschutzer *et*
*al.*? At the least, some discussion is needed about the uncertainty in Southern Ocean CO₂ flux
estimates.

-The interpolated map shown in Figure 1c is not used in the discussion and in my opinion adds nothing
to the paper that is not already shown in Figure 1b.

We embedded the subsystem boundaries to Figure 1c to outline the study region.

-Line 268: the definition used in ref 71 to determine the SACW and ESACW should be explicitly given.
We embedded the respective source water mass definitions into the method chapter.

-This may be just a matter of taste, but I disagree with the terminology of “leakage” when discussing
outgassing of natural CO₂ from the deep ocean that is associated with incomplete nutrient utilization.
Calling it a leakage implies an accidental or undesired nature, whereas the biologically-mediated
transfer of carbon to the deep ocean and the outgassing of remineralized carbon are both necessarily
in balance on a global scale, if the system is in steady state.
-The comparison to a previously published study for carbon uptake in the NBUS and SBUS on line 130
is helpful, but it would be best to mention what type of study that was (i.e., observation or model-based)
and any relevant differences.
We included further information on the reference study by stating that fluxes were observation-based
and calculated with a lesser amount of ship-board measurements as compared to our study.
-The choices of various dissociation constants to use in the carbonate system calculations (in
CO₂SYN) should be stated.
We embedded the respective dissociation constants into the method chapter.

**Reviewer #3**

This paper addresses the role of the Benguela upwelling system (BUS) in regulating the ocean uptake
of CO₂. The authors use a new compilation of shipboard measurements over the past two decades that
significantly increases the spatial/temporal coverage offered by SOCAT for pCO₂, particularly in the
coastal areas. The authors claim the BUS act as a source of CO₂ in its northern portion and as a sink
of CO₂ in its southern counterpart. The difference between the two sectors is due to a higher portion of
preformed nutrients in the upwelling water in the south with respect to the north. This results in the
biological carbon pump being more effective at decreasing surface pCO₂.
The objective of the paper are relevant for the wide climate science community and the paper has the
potential to be a significant contribution to the field. However, despite the design of the analysis seems
sound, its execution and description are very confusing with important contradictions in the parts
illustrating the main reasoning supporting the conclusions.

We thank the reviewer for acknowledging the value of our work and its potential as a significant
contribution to the scientific community.

My main concern is about lines 184-202 – This paragraph is very confusing and because it contains the
main reasoning supporting one of the highlights of the paper, it needs some improvement.

We revised this paragraph by updating the calculation and description of new production rates based
on preformed nutrients, as well as the descriptive part of the role of preformed nutrients in the BUS for
the global biological carbon pump. Further information on the specific changes we made is given below.

First of all, it uses some estimate of the volume of upwelling water which is not reported anywhere on
the paper. It should be explicitly reported in one of the tables, for example. Moreover, it starts talking
about estimating “the potential amount of CO₂ transfer into the ocean interior on the basis of new
production rates from the BUS...” but it gives an estimate of total primary production first (180-650 g
C/m²/year) for the NBUS which integrated over the area amounts to 68-245 Tg C/year. Then it gives
the actual new production estimates based on the P_{pre} share in NBUS source water reported in Table
2 (29%), which results in 19.7-71.1 Tg C/year. Now, it moves to SBUS and it gives directly the estimate
of new production without going through the calculations like it did for NBUS. This estimate is 2 orders
of magnitude lower than for NBUS. This is weird because nutrient concentration, preformed content
and area do not justify such reduction with respect to NBUS, so I assume it must be the volume of
upwelling water. Therefore, this should be explicitly reported in one of the tables.

In our revised manuscript, we state the volume of upwelling waters for both the NBUS (0.9 Sverdrup)
and SBUS (0.4 Sverdrup), while the latter is an updated estimate given by Bordbar *et al.*¹⁴. To assess
the contribution of preformed nutrients to new production, we multiplied the volume of upwelling waters
with corresponding nutrient inventories of the source water masses that are given in Supplementary
Table 4. Instead of using preformed Phosphate, we thereby decided to perform the calculations using
preformed Nitrate, as it is more likely to be the limiting factor for biological production in the BUS⁹. After
conversion to Carbon using the Redfield ratio (106:16), the resulting estimates based on preformed
nitrate amount to 14.5 Tg C year⁻¹ for the NBUS, and 7.5 Tg C year⁻¹ for the SBUS. These values imply
a higher preformed nutrient contribution in the NBUS due to the difference in the volume of upwelling
waters between the NBUS and SBUS.

We further estimated the preformed-based new production by using previously published new
production rates and the contribution of preformed nitrate (N_{pref}) as derived from our water mass
characteristics (Supplementary Table 4) to give a more robust estimate on the role of the BUS in the
carbon cycle. Hereby, the published new production rates fall within a comparatively wide range of 68 –
245 T C year⁻¹ for the NBUS and 13.5 – 42 Tg C year⁻¹ for the SBUS¹⁰⁻¹², which in sum covers
published new production rates of 241 Tg C year⁻¹ derived for the entire BUS from 16°S to 34°S¹³.
Given the contribution of preformed nutrients of 24% in the NBUS and 38% in the SBUS amounts to a
CO₂ uptake by N_{pref} utilization of 16.32 – 58.8 Tg C year⁻¹ for the NBUS, and 5.13 – 16.0 Tg C year⁻¹ for
the SBUS.

In the end, given the lower (14.5 + 7.5) and upper (58.8 + 16.0) estimates, we calculated a preformed-
based new production of ~22 – 75 Tg C year⁻¹. This implies that the biological carbon pump in the BUS
has the potential to countervail 20 up to 68% of the CO₂ release from the biological carbon pump within
the Atlantic sector of the Southern Ocean between 44° and 58°S (Fig. 4). Even if one ignores the high
estimates of the preformed-based new production while also taking into account the variability in
Southern Ocean carbon flux rates, our results emphasize the role of the BUS as a significant hub for
restoring the CO₂ uptake efficiency of the biological carbon pump.

It also states “The contribution of P_{pref} to new production from SBUS is low” but the content of P_{pref}
reported in Table 2 for this region is around 47% (i.e. way higher than for NBUS). Then it goes on
suggesting that the new production in the NBUS (19.7 – 71.1 Tg C/year) is the one that countervails
over 20% of the CO₂ loss in the subpolar South Atlantic region. This means that the compensation over
the South Atlantic CO₂ loss happens through the NBUS rather than the SBUS, which is the opposite of
what is stated throughout the paper. So, overall this reasoning appears confusing and it should be
better explained before considering this paper for publication.

Considering the difference in the volume of upwelling waters, the contribution of preformed nitrate to
new production from the SBUS is ~38%, and can therefore be considered high, and similarly, its
contribution to compensate for the CO₂ loss in the Atlantic sector of the Southern Ocean. To bring the
results of our pCO₂ recalculations and new production rates into perspective, we can say that the role
of preformed nutrients in the BUS is twofold. On the one hand, they can impact the air-sea gas
exchange by decreasing pCO₂ and fostering the drawdown of atmospheric CO₂ on a regional level. On
the other hand, from a global perspective, the use of preformed nutrients for new production in places
such as the BUS can compensate for the missed opportunity in CO₂ sequestration in high latitudes.

Further comments:

Lines 42-43. Not clear what “their formation refers to”. Is it the formation of source waters in the
Southern Ocean? Or the formation of upwelling source waters? The formation of preformed nutrients?
Consider re-phrasing.

We re-phrased the sentence to make it clear that “their formation” is referring to the formation of
preformed nutrients.

Lines 64-69. The Atlantic sector is said to be responsible of 34% of the natural CO₂ release (~110 Tg
C/year). I understand this is 34% of the total natural CO₂ release from the Southern Ocean, which is
reported here to be about 400 Tg C/year. So, isn't it just about 27.5% ?

We apologize for this mistake and made the according changes as suggested.

In Figure 1 it would be useful to see the limits defined between SBUS and NBUS.

We made the according changes as suggested.

Lines 134 – Perhaps give this number in Tg C /year for comparison.

We made the according changes as suggested.

Lines 168-169 – The calculated pCO₂ for NBUS is 50 uatm lower than the one estimated from field
measurements. This is non-negligible (11.4%) as it would make the NBUS a weaker source of CO₂. It
would be interesting if the authors could explore what are the possible reasons for this discrepancy.

We have recalculated the pCO₂ for both subsystems on the basis of the revised average source water
mass characteristics as outlined in Supplementary Table 4, resulting in a high pCO₂ in the freshly
upwelled water near the coast (NBUS: 842 – 1110 μatm, SBUS: 596 – 795 μatm) that decreases
towards the outer boundaries of the subsystems where nutrients are consumed (NBUS: 319 – 423
μatm, SBUS: 308 – 424 μatm). At the outer boundaries, measured values fall within these ranges
(Fig.3), while in both subsystems, the mean calculated pCO₂ in the freshly upwelled water exceeds the
average of our nearshore recorded pCO₂ by ~310 μatm. This difference is expected since upwelling
and the biologically-mediated drawdown of CO₂ are processes that often occur simultaneously as
indicated by e.g., satellite-derived chlorophyll concentrations showing, similar to our pCO₂ data (Fig.3),
the highest values in a narrow belt along the coast (see *Ref.*¹⁵⁻¹⁷).

The choice of figures to include in the main text is not very representative of the main results. I would
consider substituting one of the two figures with one currently in supplementary that would better guide
the reader through the reasoning of the paper (either S2 or S3).

We changed the appearance of Figures in the manuscript and included the former Supplementary
Figure 2 and 3 to the main text.

References

- 1 Wanninkhof, R. Relationship between wind speed and gas exchange over the ocean. *Journal of*
 *Geophysical Research: Oceans* **97**, 7373-7382 (1992).
- 2 Wanninkhof, R. Relationship between wind speed and gas exchange over the ocean revisited. *Limnology*
 *and Oceanography: Methods* **12**, 351-362 (2014).
- 3 Nightingale, P. D. *et al.* In situ evaluation of air-sea gas exchange parameterizations using novel
 conservative and volatile tracers. *Global Biogeochemical Cycles* **14**, 373-387 (2000).
- 4 Ho, D. T. *et al.* Measurements of air-sea gas exchange at high wind speeds in the Southern Ocean:
 Implications for global parameterizations. *Geophysical Research Letters* **33** (2006).
- 5 Wanninkhof, R. & McGillis, W. R. A cubic relationship between air-sea CO₂ exchange and wind speed.
 *Geophysical Research Letters* **26**, 1889-1892 (1999).
- 6 McCartney, M. S. Subantarctic Mode Water A Voyage of Discovery. *George Deacon 70th Anniversary*
 *Volume*, 103-119 (1977).
- 7 Karstensen, J. & Quadfasel, D. Formation of Southern Hemisphere Thermocline Waters: Water Mass
 Conversion and Subduction. *Journal of Physical Oceanography* **32**, 3020-3038 (2002).
- 8 Souza, A., Kerr, R. & Azevedo, J. On the influence of Subtropical Mode Water on the South Atlantic
 Ocean. *Journal of Marine Systems* **185** (2018).
- 9 Flohr, A., Van der Plas, A., Emeis, K., Mohrholz, V. & Rixen, T. Spatio-temporal patterns of C : N : P
 ratios in the northern Benguela upwelling regime. *Biogeosciences* **11**, 885-897 (2014).
- 10 Emeis, K. *et al.* Biogeochemical processes and turnover rates in the Northern Benguela Upwelling
 System. *Journal of Marine Systems* **188**, 63-80 (2018).
- 11 Waldron, H. N., Monteiro, P. M. S. & Swart, N. C. Carbon export and sequestration in the southern
 Benguela upwelling system: lower and upper estimates. *Ocean Science* **5**, 711-718 (2009).
- 12 Monteiro, P. M. S. in *Carbon and Nutrient Fluxes in the Continental Margins: A Global Synthesis* (eds
 458 K.K. Liu, L. Atkinson, R. Quiñones, & L. Talaue-McManus) (Springer, 2009).
- 13 Messié, M. *et al.* Potential new production estimates in four eastern boundary upwelling ecosystems.
 *Progress In Oceanography* **83**, 151-158 (2009).
- 14 Bordbar, M. H., Mohrholz, V. & Schmidt, M. The Relation of Wind-Driven Coastal and Offshore
 Upwelling in the Benguela Upwelling System. *Journal of Physical Oceanography* **51**, 3117-3133 (2021).
- 15 Weeks, S. J., Barlow, R., Roy, C. & Shillington, F. A. Remotely sensed variability of temperature and
 chlorophyll in the southern Benguela: upwelling frequency and phytoplankton response. *African Journal*
 *of Marine Science* **28**, 493-509 (2006).
- 16 Lamont, T., Barlow, R. G. & Brewin, R. J. W. Long-Term Trends in Phytoplankton Chlorophyll a and
 Size Structure in the Benguela Upwelling System. *Journal of Geophysical Research: Oceans* **124**, 1170-
 1195 (2019).
- 17 Demarcq, H., Barlow, R. & Hutchings, L. Application of a chlorophyll index derived from satellite data
 to investigate the variability of phytoplankton in the Benguela ecosystem. *African Journal of Marine*
 *Science* **29**, 271-282 (2007).

Reviewer #3 (Remarks to the Author):

This is a second review. The manuscript has improved considerably and most of the issues raised by this reviewer were addressed satisfactorily. I still have a few minor suggestions:

Lines 41-42 (Abstract): This sentence requires some thinking before making any sense. The formation of preformed nutrients does not increase directly pCO₂ in surface waters. It rather represents an inefficiency in the biological carbon pump, so this sentence is true only in relative terms – i.e. formation of preformed nutrients represent an increase in surface pCO₂ with respect to a hypothetical alternative situation where the biological carbon pump is more efficient. - I suggest to re-phrase this sentence along these lines.

Lines 143: 145: Here it says that these estimates are higher (40% and 110%) than those given above but it is actually the opposite. Consider giving more details about the meaning of these “initial” estimates and re-phrase this sentence.

Lines 162: 170: This paragraph seems to belong more to methods. I’d rather use this space to give more explanations about the relevance of these results for the broad climate science community.

Reviewer #4 (Remarks to the Author):

This study is a timely compilation and analysis of a cumulative ship-based data sets for ocean carbonate and pCO₂ in both the Northern (nBUS) and Southern Benguela Upwelling System (sBUS). In brief, this study aimed to provide top down (air-sea fluxes) and bottom up (new production fluxes) approaches to constrain the carbon budget in the two parts of the BUS. It then proceeds to use part of the analysis (New Production fluxes) to advance the importance of the role of the BUS in closing a significant part ($\pm 20\%$) of the natural CO₂ outgassing budget in the Southern Ocean south of the Polar Front, which arises from the incomplete uptake of nitrate by the iron limited new production in the Sub-Antarctic Zone.

This could be an important contribution to the widely recognized gap of coastal systems generally and upwelling systems specifically to the global ocean carbon budget, its variability and its sensitivity to climate change. However, in its present form I would not recommend it for publication in Nature.

There are three main issues: firstly, while the study presents new CO₂ observations for the region, its findings are not new (Gregor et al., 2013; Waldron et al., 2009; Monteiro, 2009; Santana Casiano et al., 2009). Perhaps most importantly, apart from mentioning the important seasonal sampling bias, it largely fails to fully discuss the implications of not resolving the spatial and temporal variability and sampling biases on the uncertainties of its constraints. Secondly, the study has 2 components – the CO₂ flux budgets and the new production fluxes – but little or no attempt is made to reconcile or discuss the interesting differences between them as well as with other earlier comparable studies in the BUS and other Eastern Boundary Upwelling Systems. This would have gone a long way to addressing the challenge posed by the title. Thirdly, the Southern Ocean angle seems an add-on to stretch the global significance of the findings. I found it lacking in process analysis which might have justified its inclusion.

Overall, the authors have done a good job of creating a climatology of the pCO₂ and FCO₂ observations for both sectors of the BUS and it is nice to see the magnitude of the constraints that have emerged from the spatial and temporal averaging in the study, relative to earlier studies. It would have benefited from a more explicit analysis of these

different approaches. Much care went into the data interpolations. However, it is not providing anything new from both a budget and process perspectives. It re-enforces what had already been concluded a decade earlier (Gregor et al., 2013; Waldron et al., 2009; Monteiro, 2009; Santana-Casiano et al., 2009) that, from a carbon budget perspective, the sBUS is a small sink and the nBUS is a significant source.

Part of the difficulty the authors may have faced is that although they had a rich data set on a decadal scale it required a spatial and temporal averaging with attendant assumptions whose consequences and impact on the uncertainties of the constraints were not adequately addressed in the context of the significant complexities in the oceanography of the BUS. Here, for example, I am referring to the implications of the well-recognized synoptic mode of variability in the sBUS whose amplitude is as large as the seasonal cycle in contrast to the nBUS where the seasonal modes are strongly linked to 2 types of mode waters. The authors furthermore link the outer boundary of the upwelling to a standard deviation threshold which they link to the influence of upwelling. What role do the authors think that mesoscale eddies might play there? The carbon biogeochemical flux calculations are limited to new production without any discussion of the role that remineralization may play in offsetting the export flux on a shelf system and weakening the mean annual flux. What do the authors think may be the implications of the seasonal bias in the data sets?

Another consideration is that while models may be weak in the context of budget criteria, they can still be very useful to test the significance of the sampling and averaging assumptions relative to the spatial and temporal scales of variability and provide a process basis for the analysis. This is a gap in the study.

Finally, while the CO₂ part of the study is clearly written, the biogeochemical section of the paper is often unclear and confusing. This should be addressed by the authors.

Response letter to the referees' comments

We hereby respond to the various comments and concerns raised during the 2nd peer review process of our manuscript entitled "Regional and global impact of CO₂ uptake in the Benguela Upwelling System through preformed nutrients", which we revised as outlined below.

Reviewer #3

This is a second review. The manuscript has improved considerably and most of the issues raised by this reviewer were addressed satisfactorily. I still have a few minor suggestions:

Lines 41-42 (Abstract): This sentence requires some thinking before making any sense. The formation of preformed nutrients does not increase directly pCO₂ in surface waters. It rather represents an inefficiency in the biological carbon pump, so this sentence is true only in relative terms – i.e. formation of preformed nutrients represent an increase in surface pCO₂ with respect to a hypothetical alternative situation where the biological carbon pump is more efficient. - I suggest to re-phrase this sentence along these lines.

We thank the reviewer for this comment and changed the sentence in the abstract to the following: "[...] Vice versa, inefficient nutrient utilization leads preformed nutrient formation, increasing pCO₂ and counteracting human-induced CO₂ invasion in the Southern Ocean. However, preformed nutrient utilization in the BUS [...]".

Lines 143: 145: Here it says that these estimates are higher (40% and 110%) than those given above but it is actually the opposite. Consider giving more details about the meaning of these "initial" estimates and re-phrase this sentence.

We thank the reviewer for this comment and made the following changes (Lines 141-144): "[...] However, in comparison to area-integrated CO₂ fluxes in ref.³⁰ for the NBUS (11.5 Tg C year⁻¹) and SBUS (-1.4 Tg C year⁻¹), our respective estimates of 15.64 (NBUS) and -2.94 Tg C year⁻¹ (SBUS) are about 40% and 110% higher, mainly due to the use of [...]".

Lines 162: 170: This paragraph seems to belong more to methods. I'd rather use this space to give more explanations about the relevance of these results for the broad climate science community.

We thank the reviewer for this comment and moved the technical part into the method section and explained our approach in more detail as follows (see Lines 158-174):

"[...] In a first step, we calculated the potential pCO₂ in freshly upwelled waters based on temperature, salinity as well as TA, DIC, and nutrient concentrations of the upwelling source water masses (ESACW and SACW, Supplementary Table 4). In a second step, effects of nutrient utilization on pCO₂ in surface waters were estimated by taken into account that phytoplankton consumes upwelled nutrients to fix DIC into biomass as also indicated by data showing nutrient depletion of offshore flowing upwelled waters at distances of about 180– 200 km to the coast^{49,50}. The Redfield ratio (106:16) is, in turn, used to translate nutrient utilization into DIC consumption and the associated release of total alkalinity (see e.g.^{1,47}). By subtracting the amount of DIC which is transformed into organic matter from the original source water DIC concentration, and adding the released TA to the original source water TA, we can calculate the pCO₂ in upwelled water after upwelled nutrients have been consumed. During the offshore flow, the upwelled water warms up as indicated by the difference between temperatures of the source waters and the sea's surface temperature. Hence, by using sea surface temperatures and

46 salinities instead of those of the source water, we can further consider the warming of upwelled waters
and its effect on pCO₂.

The exercise implies, in line with our data (Fig. 2a,b), a high pCO₂ in the freshly upwelled water near
the coast (first step: NBUS: 842 – 1110 μatm, SBUS: 596 – 795 μatm) that decreases towards the
outer boundaries of the subsystems where nutrients are consumed (third step: NBUS: 319 – 423 μatm,
SBUS: 310 – 426 μatm, Fig. 3). [...]"

**Reviewer #4**

This study is a timely compilation and analysis of a cumulative ship-based data sets for ocean
carbonate and pCO₂ in both the Northern (nBUS) and Southern Benguela Upwelling System (sBUS). In
brief, this study aimed to provide top down (air-sea fluxes) and bottom up (new production fluxes)
approaches to constrain the carbon budget in the two parts of the BUS. It then proceeds to use part of
the analysis (New Production fluxes) to advance the importance of the role of the BUS in closing a
significant part (±20%) of the natural CO₂ outgassing budget in the Southern Ocean south of the Polar
Front, which arises from the incomplete uptake of nitrate by the iron limited new production in the Sub-
Antarctic Zone. This could be an important contribution to the widely recognized gap of coastal systems
generally and upwelling systems specifically to the global ocean carbon budget, its variability and its
sensitivity to climate change. However, in its present form I would not recommend it for publication in
Nature.

We sincerely thank the reviewer for noting the novelty of this study and for providing feedback on points
which we have addressed in detail below. However, the main raised critics appear to be based on a
misunderstanding regarding the role that regenerated and preformed nutrients play in the biological
carbon pump, which seems to be a consequence of our biogeochemical section. As pointed out by the
reviewer, the CO₂ part of the study is clearly written, while the biogeochemical section seems to be
unclear and confusing at times. This issue was previously raised by reviewer #3. Hence, we have
revised this part of our manuscript to the satisfaction of reviewer #3, who suggested only minor
corrections which we have addressed as pointed out below in detail (here: Line 254-276). We hope to
have now clarified this section of our manuscript.

There are three main issues: firstly, while the study presents new CO₂ observations for the region, its
findings are not new (Gregor et al., 2013; Waldron et al., 2009; Monteiro, 2009; Santana Casiano et al.,
2009). Perhaps most importantly, apart from mentioning the important seasonal sampling bias, it largely
fails to fully discuss the implications of not resolving the spatial and temporal variability and sampling
biases on the uncertainties of its constraints.

Secondly, the study has 2 components – the CO₂ flux budgets and the new production fluxes – but little
or no attempt is made to reconcile or discuss the interesting differences between them as well as with
other earlier comparable studies in the BUS and other Eastern Boundary Upwelling Systems. This
would have gone a long way to addressing the challenge posed by the title.

Thirdly, the Southern Ocean angle seems an add-on to stretch the global significance of the findings. I
found it lacking in process analysis which might have justified its inclusion.

To issue 1: It is correct, we increased the number of direct pCO₂ observations and established a robust
climatology of pCO₂ for the BUS. Our climatology is the only one incorporating direct pCO₂
measurements along the Namibian and the South African coast within the most intensive and high
productive upwelling region. In contrast, Santana-Casiano et al. (2009) and González-Dávila et al.
(2009) measured pCO₂ along the VOS line which runs further offshore and misses the high productive
areas along the coast, as clearly shown on their first figure. As for our data set, we integrated the VOS
line data which fall into the region where low pCO₂ variability indicated a diminished influence of
upwelling. Monteiro (2009) and the follow up paper of Waldron et al. (2009) established, in turn, a

carbon budget and used the data of Santana-Casiano et al. (2009) for comparison (see below), while
Gregor et al. (2013) and Monteiro (2009) measured DIC and TA along one single transect in the SBUS
to calculate pCO₂. However, the notion of the NBUS and SBUS acting as a CO₂ source and sink,
respectively, is not new. It was postulated by Santana-Casiano et al. (2009), supported by field data
including our own (Emeis et al. 2018) and numerical model results (e.g. Brady et al., 2019).
Considering that also model studies came to this conclusion, we see no issues related to resolving the
spatial and temporal variability and sampling biases, which we have addressed in our manuscript
(Lines 124-129) and spatio-temporal interpolation method. However, what is new and what has been
acknowledged by the reviewers is that we provided additional data and followed a new notion to explain
the opposing behavior of the two systems and the BUS's role for the biological carbon pump by means
of preformed nutrient utilization. The underlying process-understanding that results from our study
thereby leads to far reaching conclusions regarding the role of Eastern Boundary Upwelling Systems as
a CO₂ sink that balances CO₂ losses at sites where preformed nutrients are formed, shedding new light
on the BUS from a budgeting and process perspective.

To issue 2: This can be divided into two parts, namely a) a discussion of earlier comparable studies in
the BUS and other Eastern Boundary Upwelling Systems, and b) a discussion of the differences
between CO₂ flux budgets and new production fluxes, both of which were considered in the manuscript
as pointed out in the following:

a) In view of earlier comparable studies in the BUS, both Monteiro (2009) and a follow up work by
Waldron et al. (2009) established a carbon budget for the BUS with the inclusion of new production
rates, and compared carbon losses from the BUS with CO₂ invasion rates as derived from Santana-
Casiano et al. (2009). As for our study, we followed this approach and compared new production rates
obtained from Waldron et al. (2009), Emeis et al. (2018) and Monteiro (2009) with CO₂ fluxes as
derived from our dataset which includes data from Santana-Casiano et al. (2009) and extensive
recordings representative for the coastal and shelf areas along the continental margin off Namibia and
South Africa. However, in contrast to previous studies from the BUS, we included impacts of the
solubility pump and differentiated between new production driven the utilization of preformed and
regenerated nutrients. A similar approach to elucidate the impact of preformed nutrients on carbon
fluxes had already been applied to the Oregon upwelling region in the California Current System (Hales
et al., 2005), as was mentioned in the Introduction of our manuscript. Hence, all previously published
data from the BUS on pCO₂ characteristics and new production, as well as concepts developed from
other EBUS that are of relevance to assess carbon budgets and the role of preformed nutrients, were
included into our study.

b) In our discussion on the difference between CO₂ flux budgets and new production fluxes (Lines 185-
196), we included impacts of the solubility pump and differentiated between new production driven by
the supply of regenerated and preformed nutrients. This is new in terms of the BUS and crucial
because of the different roles these nutrients play for the biological carbon pump, which enabled us to
explain the opposing behavior of the two subsystems.

To issue 3: Since we applied the well-known influence of preformed nutrients on the CO₂ uptake of the
biological carbon pump and the central role deep water formation in the Southern Ocean plays for the
cycling of preformed nutrients to the BUS, omitting the Southern Ocean would have only led to
misinterpretations of the results derived from the BUS. For instance, we showed that the biological
carbon pump takes up CO₂ in the BUS via the utilization of preformed nutrients. Ignoring that the
biological carbon pump loses CO₂ during the formation of preformed nutrients in the Southern Ocean
would have led to the impression that the BUS acts as a net CO₂ uptake region, whereas in steady
state, the CO₂ uptake through the utilization of preformed nutrients balances the CO₂ loss caused

during their formation. The latter is thereby illustrated in Figure 4, showing the cycling of preformed
nutrients and their contribution to new production in the BUS.

Overall, the authors have done a good job of creating a climatology of the pCO₂ and FCO₂ observations
for both sectors of the BUS and it is nice to see the magnitude of the constraints that have emerged
from the spatial and temporal averaging in the study, relative to earlier studies. It would have benefited
from a more explicit analysis of these different approaches.

We thank the reviewer for acknowledging our approach of handling the pCO₂ and shipboard data for
spatio-temporal interpolations. As pointed out before, this is the only study which presents a pCO₂
climatology with direct measurement of pCO₂ within the key upwelling regions that also includes data
which we have previously published (Emeis et al., 2019). Within this study, we expanded our existing
data set and applied - in cooperation with Peter Landschützer and due to the recommendation of
reviewer #2 - a new interpolation scheme which we described within the method section. As pointed out
in our manuscript, we incorporated pCO₂ records gathered during 14 cruises to the BUS and
embedded quality-controlled measurements from the Surface Ocean CO₂ Atlas (SOCAT) v2020 into
our analysis, resulting in a dataset spanning a timeframe from 1986 to 2020 with over 250 000 data
points. This extended dataset includes also data from the VOS line, that has been presented in earlier
work (Santana-Casiano et al., 2009; González-Dávila et al., 2009), while excluding pCO₂ data derived
from DIC and TA measurements (Gregor et al., 2013; Monteiro, 2009). In addition, our climatology
differs from previously published pCO₂ climatologies (e.g. Laruelle et al., 2017; Landschützer et al.,
2020), as ours is not solely based on SOCAT data, which misses coverage particularly in the NBUS
coastal region and hence major upwelling impacts on pCO₂. To emphasize the novelty of our study with
respect to previous work, we included the following statement into our manuscript:

Lines 272-276: “[...] The uncertainty in average estimates thereby provides an outline on the strong
variability of pCO₂ that can be found in coastal upwelling settings. In addition, our pCO₂ climatology
offers an updated view on CO₂ sources and sinks in comparison to previous pCO₂ climatologies^{1,2},
which were merely based on the SOCAT dataset that largely misses pCO₂ recordings in the NBUS
coastal region (Fig.1a) [...]”

Much care went into the data interpolations. However, it is not providing anything new from both a
budget and process perspectives. It re-enforces what had already been concluded a decade earlier
(Gregor et al., 2013; Waldron et al., 2009; Monteiro, 2009; Santana-Casiano et al., 2009) that, from a
carbon budget perspective, the sBUS is a small sink and the nBUS is a significant source.

We agree with the above concerning the NBUS source and SBUS sink budgeting. Nevertheless, we
updated previous pCO₂ climatologies by including additional pCO₂ data and a new interpolation
scheme, but, as also mentioned before, the main objective of this work is to explain the opposing
behaviour of the two subsystems by means of preformed nutrient utilization.

Part of the difficulty the authors may have faced is that although they had a rich data set on a decadal
scale it required a spatial and temporal averaging with attendant assumptions whose consequences
and impact on the uncertainties of the constraints were not adequately addressed in the context of the
significant complexities in the oceanography of the BUS. Here, for example, I am referring to the
implications of the well-recognized synoptic mode of variability in the sBUS whose amplitude is as large
as the seasonal cycle in contrast to the nBUS where the seasonal modes and strongly linked to 2 types
of mode waters.

We thank the reviewer for noting the richness in data on which our work is based on. Indeed, there are
several uncertainties, but this does not affect the main conclusion that the SBUS is a small sink and the
NBUS a significant source, which, in turn, poses the question as to why that is the case. However, the

variability in source water masses (SACW, ESACW) could affect the concentration of preformed
nutrients that are being upwelled into the surface region, and with it, the efficiency of the biological
carbon pump. To address the impact of source water mass variabilities on CO₂ fluxes, we utilized the
average source water mass characteristics (which includes DIC, TA, nutrients, temperature, salinity) to
simulate sea surface pCO₂ together with its thermal and biological controls as outlined in the
biogeochemical section. We derived the average source water mass characteristics on the basis of
many different cruises during the past 2 decades as outlined in Supplementary Table 4, including data
from the Global Ocean Data Analysis Project version 2.2020 (GLODAPv2_2020). By also taking into
account the standard error of our average source water mass calculations, we were able to include the
impact of SACW/ESACW variability into our analysis, which is also mirrored in the error bars in Figure
3. As for the impact of SACW/ESACW variability on new production, we applied two approaches based
on upwelling velocities obtained from a model study (Bordbar et al., 2021) which integrates spatial and
temporal variabilities in upwelling intensities, and new production rates previously published by Emeis
et al. (2018), Monteiro (2009) and Waldron et al. (2009). As for our first approach, upwelling velocities
for the NBUS and SBUS were multiplied by average performed nutrient concentrations as derived from
our source water mass characteristics. For the second approach, we estimated new production rates by
using the contribution of preformed nutrients to the total nutrient concentrations. For both approaches,
we included the standard errors of the preformed nutrient concentrations. Given these different
approaches and the rich database our average source water mass characteristics are based upon, we
have embedded the impact of source water mass variabilities into our manuscript, and have addressed
them accordingly by outlining the uncertainties in pCO₂ simulations as well as new production rates.

The authors furthermore link the outer boundary of the upwelling to a standard deviation threshold
which they link to the influence of upwelling. What role do the authors think that mesoscale eddies
might play there?

Our dataset consists of various cruise underway pCO₂ records taken across the shelf area, which
includes small- and mesoscale variabilities in pCO₂ as implied by the amplitude in the standard
deviation at the coast and further offshore (Fig. 2a,b).

The carbon biogeochemical flux calculations are limited to new production without any discussion of the
role that remineralization may play in offsetting the export flux on a shelf system and weakening the
mean annual flux.

Here, the reviewer is likely referring to our pCO₂ simulations, where we accounted for the effects of
upwelling, surface warming and photosynthesis on sea surface pCO₂. However, we also took into
account the regenerated nutrient component in the nutrient budget that upwells to the surface, which
resembles the pathway “B – Shelf re-cycled nitrate” in Waldron et al. (2009), and hence the
remineralization process on the shelf. Hereby, the effect of remineralization on DIC and TA on
upwelling waters has also been included in our calculations, as we used the DIC and TA source water
mass concentrations measured across the in- and offshore shelf system, where remineralization takes
place.

What do the authors think may be the implications of the seasonal bias in the data sets?

Considering that also model results capture the opposing function of the SBUS and NBUS as a CO₂
sink and source, respectively, we are quite confident that a seasonal bias in the data sets is neither
affecting this result, nor the resulting main question as to what are underlying processes explaining the
opposing behaviour of the two subsystems.

Another consideration is that while models may be weak in the context of budget criteria, they can still
be very useful to test the significance of the sampling and averaging assumptions relative to the spatial

and temporal scales of variability and provide a process basis for the analysis. This is a gap in the
study.

Yes, we fully agree. As mentioned before, model results principally agree to observations, showing the
opposing functions of SBUS and NBUS to be robust, but they are still prone to model deficiencies as
pointed out by the authors themselves (see e.g. Brady et al., 2019: "The BenCS has larger physical
biases in the CESM-LENS than all other EBUSs. [...] This bias is likely driven by the fact that the
Angola-Benguela front is simulated too far south, in addition to deficiencies in upwelling and meridional
transport that are caused by unrealistic alongshore wind stress structure [...]"). Hereby, our established
climatology of pCO₂ is, in turn, quite important to further constrain model results and to guide the model
evaluation by embodying coastal pCO₂ variabilities that are largely missing in SOCAT-based pCO₂
products (see Fig. 1a), and by emphasizing the role of performed nutrients.

Finally, while the CO₂ part of the study is clearly written, the biogeochemical section of the paper is
often unclear and confusing. This should be addressed by the authors.

We thank the reviewer for this feedback and would like to apologize for any confusions our writing has
led to. Based on the comments we received from the first peer-review process, we have restructured
this section to the best of our abilities to avoid unclear and confusing statements, and addressed further
suggestions of this part of our manuscript as raised by reviewer #3 in the following manner:

Lines 158-174: "[...] In a first step, we calculated the potential pCO₂ in freshly upwelled waters based
on temperature, salinity as well as TA, DIC, and nutrient concentrations of the upwelling source water
masses (ESACW and SACW, Supplementary Table 4). In a second step, effects of nutrient utilization
on pCO₂ in surface waters were estimated by taken into account that phytoplankton consumes
upwelled nutrients to fix DIC into biomass as also indicated by data showing nutrient depletion of
offshore flowing upwelled waters at distances of about 180– 200 km to the coast^{49,50}. The Redfield ratio
(106:16) is, in turn, used to translate nutrient utilization into DIC consumption and the associated
release of total alkalinity (see e.g.^{1,47}). By subtracting the amount of DIC which is transformed into
organic matter from the original source water DIC concentration, and adding the released TA to the
original source water TA, we can calculate the pCO₂ in upwelled water after upwelled nutrients have
been consumed. During the offshore flow, the upwelled water warms up as indicated by the difference
between temperatures of the source waters and the sea's surface temperature. Hence, by using sea
surface temperatures and salinities instead of those of the source water, we can further consider the
warming of upwelled waters and its effect on pCO₂.

The exercise implies, in line with our data (Fig. 2a,b), a high pCO₂ in the freshly upwelled water near
the coast (first step: NBUS: 842 – 1110 μatm, SBUS: 596 – 795 μatm) that decreases towards the
outer boundaries of the subsystems where nutrients are consumed (third step: NBUS: 319 – 423 μatm,
SBUS: 310 – 426 μatm, Fig. 3). [...]"

Bordbar, Mohammad Hadi, Volker Mohrholz, and Martin Schmidt. "The Relation of Wind-Driven Coastal and
Offshore Upwelling in the Benguela Upwelling System." [In English]. *Journal of Physical Oceanography*
51, no. 10 (01 Oct. 2021 2021): 3117-33. <https://doi.org/10.1175/JPO-D-20-0297.1>.
<https://journals.ametsoc.org/view/journals/phoc/51/10/JPO-D-20-0297.1.xml>.

Brady, Riley, Nicole Lovenduski, Michael Alexander, Michael Jacox, and Nicolas Gruber. "On the Role of
Climate Modes in Modulating the Air-Sea Co₂ Fluxes in Eastern Boundary Upwelling Systems."
*Biogeosciences* 16 (September 21, 2019 2019): 329-46. <https://doi.org/10.5194/bg-2018-415>.

Emeis, Kay, Anja Eggert, Anita Flohr, Niko Lahajnar, Günther Nausch, Andreas Neumann, Tim Rixen, *et al.*
"Biogeochemical Processes and Turnover Rates in the Northern Benguela Upwelling System." *Journal of*
*Marine Systems* 188 (2018 2018): 63-80.
González-Dávila, Melchor, J. Magdalena Santana-Casiano, and Ivan R. Ucha. "Seasonal Variability of Fco2 in
the Angola-Benguela Region." *Eastern Boundary Upwelling Ecosystems: Integrative and Comparative*
*Approaches* 83, no. 1 (December 1, 2009 2009): 124-33. <https://doi.org/10.1016/j.pocean.2009.07.033>.
Gregor, Luke, and P.M.S. Monteiro. "Is the Southern Benguela a Significant Regional Sink of Co2?". *South*
*African Journal of Science* 109 (2013 2013).
Hales, Burke, Taro Takahashi, and Leah Bandstra. "Atmospheric Co2 Uptake by a Coastal Upwelling System."
*Global Biogeochemical Cycles* 19, no. 1 (March 1, 2005 2005). <https://doi.org/10.1029/2004GB002295>.
Landschützer, Peter, Goulven Laruelle, Alizee Roobaert, and Pierre Regnier. "A Uniform P Co2 Climatology
Combining Open and Coastal Oceans." *Earth Syst. Sci. Data* (May 6, 2020 2020).
Laruelle, Goulven G., Ronny Lauerwald, Benjamin Pfeil, and Pierre Regnier. "Regionalized Global Budget of the
Co2 Exchange at the Air-Water Interface in Continental Shelf Seas." *Global Biogeochemical Cycles* 28,
no. 11 (November 1, 2014 2014): 1199-214. <https://doi.org/10.1002/2014GB004832>.
Monteiro, P. M. S. "Carbon Fluxes in the Benguela Upwelling System." In *Carbon and Nutrient Fluxes in the*
*Continental Margins: A Global Synthesis*, edited by K.K. Liu, L. Atkinson, R. Quiñones and L. Talaue-
McManus. Berlin: Springer, 2009.
Santana-Casiano, J. Magdalena, Melchor González-Dávila, and Ivan R. Ucha. "Carbon Dioxide Fluxes in the
Benguela Upwelling System During Winter and Spring: A Comparison between 2005 and 2006." *Surface*
*Ocean CO2 Variability and Vulnerabilities* 56, no. 8 (April 1, 2009 2009): 533-41.
<https://doi.org/10.1016/j.dsr2.2008.12.010>.
Waldron, H.N., P.M.S. Monteiro, and N.C. Swart. "Carbon Export and Sequestration in the Southern Benguela
Upwelling System: Lower and Upper Estimates." *Ocean Science* 5 (2009 2009): 711-18.

Reviewer #4 (Remarks to the Author):

The manuscript has 2 parts. 1: calculation of the mean pCO₂ and annual fluxes of the NBUS and SBUS; 2: the calculation of the carbon export driven by the pre-formed nutrients in SAMW that supply SACW and ESACW and the finding that the pre-formed nutrients linked export flux makes a significant contribution to offsetting the outgassing flux of CO₂ from upwelled CDW in the Southern Ocean (Fig 4).

The authors have addressed some of the minor issues but there are still some fundamental structural and conceptual problems with the study. The fundamental problem is that the two parts are not connected to together inform Fig 4.

I suggest that the authors restructure the paper

Section 1: Air-Sea CO₂ fluxes in the NBUS and SBUS calculated from total NO₃-T and from pre-formed NO₃-PF (Include a Table or a diagram)

Section 2: Connect the CO₂ fluxes from NO₃-T calculation to the observed fluxes.

Include the pCO₂ decomposition (Discuss assumption and contrast to earlier work and its own assumptions)

Section 3: Connect the CO₂ fluxes from NO₃-PF calculation to the outgassing in the SO

Let me try to articulate some of the major issues and then I address their responses to my initial comments.

CO₂ fluxes:

Given that the role of preformed nutrient linked export flux in balancing the CO₂ loss in the Southern Ocean is a central part of this work as encapsulated in Fig 4, it is rather unclear why the study starts with an assessment of the CO₂ fluxes. This is especially problematic because the net CO₂ fluxes in the NBUS and SBUS are derived from total nitrate supply not just the pre-formed flux, which is the main point of the study (Fig 4). Hence, the connection to what follows is unclear.

It seems that the key point in this, apparently out of place, initial part of the paper is to highlight the net CO₂ flux differences between the NBUS and SBUS and attribute these differences to surface warming in the NBUS. While surface warming does play a role as was shown in Monteiro, 2010; Santana Casiano, 2009 and Gregor et al., 2013; 2014) it is an oversimplification without a fuller discussion of the multiple complex physics and biogeochemical feedbacks in both the NBUS and SBUS that contribute to the effectiveness of the assumption of a linear link between pre-formed nutrients and carbon export. These complexities (seasonal shelf circulation, carbon deposition on shelf, influence of shelf width on circulation and remineralization, are set out in multiple publications that explain the physics – biogeochemistry links and their seasonal dynamics in the NBUS and SBUS. So, for example would one not expect the combination of deep mixing in the inner shelf (<200m) and strong winds in winter to explain the elevated pCO₂s that would drive the strong outgassing fluxes in the NBUS (Fig 2b)?

I suggest that apart from a more thorough discussion of the assumptions, that the authors provide quantitative support for the role of temperature by doing a decomposition of the pCO₂ gradients in time and space into their thermal and non-thermal components. This would provide the required support for the conclusion that the outgassing fluxes in the NBUS are thermally controlled and in gassing in the SBUS and non-thermally controlled. The assignment of the boundary separating the NBUS and SBUS is also different to the published literature, and it has implications for the calculation of the total fluxes. A more thorough justification of this choice is needed together with the implications in comparing the fluxes from historical studies. This has implications for the following section.

Pre-formed production and export fluxes

A first issue with this section is again the geographic boundary between the SBUS and the NBUS at 26oS which means that the pre-formed nutrient fluxes of the very large Lüderitz upwelling cell are allocated to the SBUS instead of the NBUS. The assignment of boundaries in the BUS is the subject of several studies Hutchings et al., 2004; Monteiro, 2010 (from Monteiro 1996); Monteiro et al., 2012. I encourage the authors to have a careful read of these and other related papers. The choice of boundaries is important for upwelling fluxes, surface areas and interpretation of the role of the oceanographic dynamics in comparing different studies. In a 2 sector formalism the large Luderitz upwelling cell is normally assigned to the NBUS because it transports ESACW onto the NBUS shelf in winter-spring. Assigning it to the SBUS would in effect reduce the export flux in the NBUS and increase it in the SBUS, which could explain part of the asymmetry emerging from this study relative to earlier studies.

Response to earlier rebuttal:

We sincerely thank the reviewer for noting the novelty of this study and for providing feedback on points which we have addressed in detail below. However, the main raised critics appear to be based on a misunderstanding regarding the role that regenerated and preformed nutrients play in the biological carbon pump, which seems to be a consequence of our biogeochemical section. As pointed out by the reviewer, the CO₂ part of the study is clearly written, while the biogeochemical section seems to be unclear and confusing at times. This issue was previously raised by reviewer #3. Hence, we have revised this part of our manuscript to the satisfaction of reviewer #3, who suggested only minor corrections which we have addressed as pointed out below in detail (here: Line 254-276). We hope to have now clarified this section of our manuscript.

The focus on pre-formed nutrients is certainly clearer but there is a terminology problem here. The use of regenerated nutrients to refer to those remineralised in the thermocline waters is confusing because regenerated is mostly used in the context of the mixed layer where it helps define the f-ratio. The problem is that from a biogeochemical perspective regenerated nutrients do not contribute to new production that set the carbon export fluxes whereas in this case both preformed and what are called regenerated nutrients do contribute to new production and export. I suggest that the authors be more specific and refer to nutrients remineralized in the thermocline waters. This will make it clearer which fraction of the nutrient supply they are referring to. To issue 1: It is correct, we increased the number of direct pCO₂ observations and established a robust climatology of pCO₂ for the BUS. Our climatology is the only one incorporating direct pCO₂ measurements along the Namibian and the South African coast within the most intensive and high productive upwelling region. In contrast, Santana-Casiano et al. (2009) and González-Dávila et al. (2009) measured pCO₂ along the VOS line which runs further offshore and misses the high productive areas along the coast, as clearly shown on their first figure. As for our data set, we integrated the VOS line data which fall into the region where low pCO₂ variability indicated a diminished influence of upwelling. Monteiro (2010) and the follow up paper of Waldron et al. (2009) established, in turn, a carbon budget and used the data of Santana-Casiano et al. (2009) for comparison (see below), while Gregor et al. (2013) and **Monteiro (2010) measured DIC and TA along one single transect in the SBUS to calculate pCO₂. This is incorrect: the Monteiro 2010 box model is constructed from 3 ship cross shelf sections spanning the northern, central and southern Benguela upwelling sub-systems and the Gregor et al., 2013 study was based on six sections that spanned a full seasonal cycle in the SBUS.

** Monteiro 2009 is actually Monteiro 2010 – the year of publication of the book. However, the notion of the NBUS and SBUS acting as a CO₂ source and sink, respectively, is not new. Correct. (Monteiro, 1996; 2010; Santana-Casiano, 2009). Moreover the sub-system characteristics of the BUS have been well defined both in terms of upwelling centres (Monteiro 2010) and ecologically (Hutchings 2004). This is important in this study because of the biogeochemical component. It was postulated by Santana-Casiano et al. (2009), supported by field data including our own (Emeis et al. 2018) and numerical model results (e.g. Brady et al., 2019).

This is only partially correct and suggests an incomplete reading of critical background references. It was first proposed in a mechanistically consistent box-model from the temporal and spatial characteristics of wind stress and Ekman transport at each of the 6 main upwelling centres by Monteiro 1996; 2010. As the authors suggest the Santana Casiano 2009 observations in the NBUS are beyond their own boundaries of the upwelling system so the outgassing conclusion is derived mainly from thermal impact on pCO₂.

Considering that also model studies came to this conclusion, we see no issues related to resolving the spatial and temporal variability and sampling biases, which we have addressed in our manuscript (Lines 124-129) and spatio-temporal interpolation method. This comparison requires more than a cursory comment

However, what is new and what has been acknowledged by the reviewers is that we provided additional data and followed a new notion to explain the opposing behavior of the two systems and the BUS's role for the biological carbon pump by means of preformed nutrient utilization.

New data is not equivalent to new insights. I do not see the point of including the observations based pCO₂ and flux calculations, which are the outcome of both preformed and re-mineralized nutrient fluxes, in a study that is primarily aiming to constrain the carbon export from pre-formed nitrate alone. It's just an add on and it confuses the primary focus of the paper set out in Fig 4. This means that the CO₂ fluxes and carbon export production fluxes are not integrated and that detracts from the significance of the paper.

The underlying process-understanding that results from our study thereby leads to far reaching conclusions regarding the role of Eastern Boundary Upwelling Systems as a CO₂ sink that balances CO₂ losses at sites where preformed nutrients are formed, shedding new light on the BUS from a budgeting and process perspective.

As discussed below in (issue 3), this may be so but it requires a more careful discussion on the limitations of this assertion.

To issue 2: This can be divided into two parts, namely a) a discussion of earlier comparable studies in the BUS and other Eastern Boundary Upwelling Systems, and b) a discussion of the differences between CO₂ flux budgets and new production fluxes, both of which were considered in the manuscripts pointed out in the following:

a) In view of earlier comparable studies in the BUS, both Monteiro (2009) and a follow up work by Waldron et al. (2009) established a carbon budget for the BUS with the inclusion of new production rates, and compared carbon losses from the BUS with CO₂ invasion rates as derived from Santana- Casiano et al. (2009). As for our study, we followed this approach and compared new production rates obtained from Waldron et al. (2009), Emeis et al. (2018) and Monteiro (2009) with CO₂ fluxes as derived from our dataset which includes data from Santana-Casiano et al. (2009) and extensive recordings representative for the coastal and shelf areas along the continental margin off Namibia and South Africa.

However, in contrast to previous studies from the BUS, we included impacts of the solubility pump and differentiated between new production driven the utilization of preformed and regenerated nutrients.

Both Monteiro 2010 and Monteiro model in Waldron et al., 2009 correct the pCO₂ for warming of upwelled waters inshore and offshore.

A similar approach to elucidate the impact of preformed nutrients on carbon fluxes had already been applied to the Oregon upwelling region in the California Current System (Hales et al., 2005), as was mentioned in the Introduction of our manuscript. Hence, all previously published data from the BUS on pCO₂ characteristics and new production, as well as concepts developed from other EBUS that are of relevance to assess carbon budgets and the role of preformed nutrients, were included into our study.

What is missing is a more thorough discussion of the consequences of the assumptions and choices made in this study relative to historical work. This should include a table of both CO₂ fluxes and carbon export fluxes and nutrient boundary conditions from the different studies.

b) In our discussion on the difference between CO₂ flux budgets and new production fluxes (Lines 185- 196), we included impacts of the solubility pump and differentiated between new production driven by the supply of regenerated and preformed nutrients.

This is new in terms of the BUS and crucial because of the different roles these nutrients play for the biological carbon pump, which enabled us to explain the opposing behavior of the two subsystems.

I am not convinced that the last sentence logically follows from the first

To issue 3: Since we applied the well-known influence of preformed nutrients on the CO₂ uptake of the biological carbon pump and the central role deep water formation in the Southern Ocean plays for the cycling of preformed nutrients to the BUS, omitting the Southern Ocean would have only led to misinterpretations of the results derived from the BUS. For instance, we showed that the biological carbon pump takes up CO₂ in the BUS via the utilization of preformed nutrients. Ignoring that the biological carbon pump loses CO₂ during the formation of preformed nutrients in the Southern Ocean would have led to the impression that the BUS acts as a net CO₂ uptake region, whereas in steady state, the CO₂ uptake through the utilization of preformed nutrients balances the CO₂ loss caused during their formation. The latter is thereby illustrated in Figure 4, showing the cycling of preformed nutrients and their contribution to new production in the BUS.

Yes this study provides a focus on pre-formed nutrient boundary conditions to the BUS but it fails to adequately discuss the implications of this potentially interesting approach and its underlying assumptions for what is a complex system, both biogeochemically and physically. Fig 4 seems to suggest a simple link between carbon export linked to pre-formed nutrients and compensation of CO₂ outgassing in the Southern Ocean. In reality there is a complex set of physics and biogeochemical feedbacks that are well recognized and published that most likely influence the impact of these assumptions on the magnitude of the carbon export fluxes. These include shelf deposition especially in the NBUS, both aerobic and anaerobic remineralization of the exported carbon, denitrification, nitrification. I would not expect the authors to come up with a detailed study of these factors but at least discuss the contribution and uncertainty levels that they make to re-balancing the CO₂ loss in the Southern Ocean. The rather large number for the contribution by the Benguela to offsetting outgassing of CO₂ in the Southern Ocean requires stronger justification and clearer uncertainties.

Response letter to the referees' comments

We hereby respond to the various comments and concerns raised during the 3rd peer review process of our manuscript entitled "Regional and global impact of CO₂ uptake in the Benguela Upwelling System through preformed nutrients", which we revised as outlined (*in blue*) below.

Reviewer #4

The manuscript has 2 parts. 1: calculation of the mean pCO₂ and annual fluxes of the NBUS and SBUS; 2: the calculation of the carbon export driven by the pre-formed nutrients in SAMW that supply SACW and ESACW and the finding that the pre-formed nutrients linked export flux makes a significant contribution to offsetting the outgassing flux of CO₂ from upwelled CDW in the Southern Ocean (Fig 4). The authors have addressed some of the minor issues but there are still some fundamental structural and conceptual problems with the study. The fundamental problem is that the two parts are not connected to together inform Fig 4. I suggest that the authors restructure the paper:

Section 1: Air-Sea CO₂ fluxes in the NBUS and SBUS calculated from total NO₃-T and from pre-formed NO₃-PF (Include a Table or a diagram)

Section 2: Connect the CO₂ fluxes from NO₃-T calculation to the observed fluxes. Include the pCO₂ decomposition (Discuss assumption and contrast to earlier work and its own assumptions)

Section 3: Connect the CO₂ fluxes from NO₃-PF calculation to the outgassing in the SO

We thank the reviewer for noting our approach to addressing and overcoming the issues raised during the last peer review process. With the additional feedback from the editor concerning the structure of the manuscript, we will at this stage be focussing on solving the conceptual problems as explained in our responses to the following comments.

Let me try to articulate some of the major issues and then I address their responses to my initial comments.

CO₂ fluxes:

Given that the role of preformed nutrient linked export flux in balancing the CO₂ loss in the Southern Ocean is a central part of this work as encapsulated in Fig 4, it is rather unclear why the study starts with an assessment of the CO₂ fluxes. This is especially problematic because the net CO₂ fluxes in the NBUS and SBUS are derived from total nitrate supply not just the pre-formed flux, which is the main point of the study (Fig 4). Hence, the connection to what follows is unclear.

It seems that the key point in this, apparently out of place, initial part of the paper is to highlight the net CO₂ flux differences between the NBUS and SBUS and attribute these differences to surface warming in the NBUS. While surface warming does play a role as was shown in Monteiro, 2010; Santana Casiano, 2009 and Gregor et al., 2013; 2014) it is an oversimplification without a fuller discussion of the multiple complex physics and biogeochemical feedbacks in both the NBUS and SBUS that contribute to the effectiveness of the assumption of a linear link between pre-formed nutrients and carbon export. These complexities (seasonal shelf circulation, carbon deposition on shelf, influence of shelf width on circulation and remineralization, are set out in multiple publications that explain the physics – biogeochemistry links and their seasonal dynamics in the NBUS and SBUS. So, for example would one not expect the combination of deep mixing in the inner shelf (<200m) and strong winds in winter to explain the elevated pCO₂s that would drive the strong outgassing fluxes in the NBUS (Fig 2b)?

I suggest that apart from a more thorough discussion of the assumptions, that the authors provide
quantitative support for the role of temperature by doing a decomposition of the pCO₂ gradients in time
and space into their thermal and non-thermal components. This would provide the required support for
the conclusion that the outgassing fluxes in the NBUS are thermally controlled and in gassing in the
SBUS and non-thermally controlled. The assignment of the boundary separating the NBUS and SBUS
is also different to the published literature, and it has implications for the calculation of the total fluxes. A
more thorough justification of this choice is needed together with the implications in comparing the
fluxes from historical studies. This has implications for the following section.

The initial part of our manuscript is relevant to understand the role of preformed nutrients for the
regional CO₂ sources/sinks, since not only the temperature effect is crucial for pCO₂, but also the use of
preformed nutrients (as already shown e.g. by Hales et al. 2005). In order to elaborate the temperature
effect on pCO₂, we have included an analysis of (non-) thermal components in pCO₂ with an additional
Figure 4 as outlined in the main text in the section “Nutrients as a driver of regional variability in pCO₂”.
The underlying methods are thereby described in a new section called “(Non-) thermal component
analysis of sea surface pCO₂”. As for the boundary assignment for separating NBUS from SBUS, we
have addressed the issue in the next comment below to which we kindly refer (lines 76-91).

Pre-formed production and export fluxes

A first issue with this section is again the geographic boundary between the SBUS and the NBUS at
26oS which means that the pre-formed nutrient fluxes of the very large Lüderitz upwelling cell are
allocated to the SBUS instead of the NBUS. The assignment of boundaries in the BUS is the subject of
several studies Hutchings et al., 2004; Monteiro, 2010 (from Monteiro 1996); Monteiro et al., 2012. I
encourage the authors to have a careful read of these and other related papers. The choice of
boundaries is important for upwelling fluxes, surface areas and interpretation of the role of the
oceanographic dynamics in comparing different studies. In a 2 sector formalism the large Lüderitz
upwelling cell is normally assigned to the NBUS because it transports ESACW onto the NBUS shelf in
winter-spring. Assigning it to the SBUS would in effect reduce the export flux in the NBUS and increase
it in the SBUS, which could explain part of the asymmetry emerging from this study relative to earlier
studies.

We fully agree to the point raised about the importance of boundary choices in determining upwelling
fluxes and subsequent assumptions on carbon flux dynamics. As there has been less consensus on the
definition of the BUS's offshore boundary which has led to strong discrepancies in offshore boundary
assignments, we found it particularly important to include the analysis of sea surface pCO₂
characteristics as outlined in Fig.2a,b to address this issue (see manuscript line 90-100). In view of the
latitudinal boundary, we adopted the location of the Lüderitz upwelling cell at ~26°S, since it is
commonly defined as “[...] the boundary between the northern and southern sub-systems (Lüderitz:
26°S) [...]” (Monteiro et al., 2010, p.67), as also stated in e.g., Shillington et al. (2013), Hutchings et al.
(2009) and Rae (2005). We included the following lines into the main manuscript section on pCO₂
characteristics to outline this boundary choice: (lines 93-95: “The latitudinal boundary between the
NBUS and SBUS is formed by the Lüderitz cell (~26°S)³⁵⁻³⁷ within the Lüderitz upwelling region (24°S-
28°S)^{38,39} that is subject to perennial coastal upwelling.”). While there seems to be solid consensus
concerning the latitudinal separation of the two subsystems (Lüderitz cell), other studies further propose
the Lüderitz upwelling region, that impacts the air-sea gas exchange in the SBUS and NBUS across
24°S – 28°S (Santana-Casiano et al., 2009; González-Dávila et al., 2009), to be seen as a separate
zone for calculating CO₂ fluxes within the BUS.

Response to earlier rebuttal:

*"We sincerely thank the reviewer for noting the novelty of this study and for providing feedback on*
*points which we have addressed in detail below. However, the main raised critics appear to be based*
*on a misunderstanding regarding the role that regenerated and preformed nutrients play in the*
*biological carbon pump, which seems to be a consequence of our biogeochemical section. As pointed*
*out by the reviewer, the CO₂ part of the study is clearly written, while the biogeochemical section*
*seems to be unclear and confusing at times. This issue was previously raised by reviewer #3. Hence,*
*we have revised this part of our manuscript to the satisfaction of reviewer #3, who suggested only*
*minor corrections which we have addressed as pointed out below in detail (here: Line 254-276). We*
*hope to have now clarified this section of our manuscript."*

The focus on pre-formed nutrients is certainly clearer but there is a terminology problem here. The use
of regenerated nutrients to refer to those remineralised in the thermocline waters is confusing because
regenerated is mostly used in the context of the mixed layer where it helps define the f-ratio. The
problem is that from a biogeochemical perspective regenerated nutrients do not contribute to new
production that set the carbon export fluxes whereas in this case both preformed and what are called
regenerated nutrients do contribute to new production and export. I suggest that the authors be more
specific and refer to nutrients remineralized in the thermocline waters. This will make it clearer which
fraction of the nutrient supply they are referring to.

The term 'preformed' was coined by R. M. Pythkowicz and D. R. Kester (1966) and in particular by
Broecker et al. (1985), and since the benchmark paper of Ito & Follows (2005), it is directly linked to the
term 'regenerated' in order to distinguish between these two different pools of nutrients in the ocean.
We are aware that the term 'regenerated' was also introduced by Eppley and Peterson (1979) as
mentioned by the reviewer, but it is beyond the scope of this paper to change established scientific
terms.

*"To issue 1: It is correct, we increased the number of direct pCO₂ observations and established a*
*robust climatology of pCO₂ for the BUS. Our climatology is the only one incorporating direct pCO₂*
*measurements along the Namibian and the South African coast within the most intensive and high*
*productive upwelling region. In contrast, Santana-Casiano et al. (2009) and González-Dávila et al.*
*(2009) measured pCO₂ along the VOS line which runs further offshore and misses the high productive*
*areas along the coast, as clearly shown on their first figure. As for our data set, we integrated the VOS*
*line data which fall into the region where low pCO₂ variability indicated a diminished influence of*
*upwelling. Monteiro (2010) and the follow up paper of Waldron et al. (2009) established, in turn, a*
*carbon budget and used the data of Santana-Casiano et al. (2009) for comparison (see below), while*
*Gregor et al. (2013) and **Monteiro (2010) measured DIC and TA along one single transect in the*
*SBUS to calculate pCO₂."*

This is incorrect: the Monteiro 2010 box model is constructed from 3 ship cross shelf sections spanning
the northern, central and southern Benguela upwelling sub-systems and the Gregor et al., 2013 study
was based on six sections that spanned a full seasonal cycle in the SBUS.

** Monteiro 2009 is actually Monteiro 2010 – the year of publication of the book

We apologize for our mistake, and have made the according changes within our reference list.

*"However, the notion of the NBUS and SBUS acting as a CO₂ source and sink, respectively, is not*
*new."*

Correct. (Monteiro, 1996; 2010; Santana-Casiano, 2009). Moreover the sub-system characteristics of
the BUS have been well defined both in terms of upwelling centres (Monteiro 2010) and ecologically
(Hutchings 2004). This is important in this study because of the biogeochemical component.

We thank the reviewer for this comment and have integrated the missing reference into the introduction
section of our manuscript.

*"It was postulated by Santana-Casiano et al. (2009), supported by field data including our own (Emeis*
*et al. 2018) and numerical model results (e.g. Brady et al., 2019)."*

This is only partially correct and suggests an incomplete reading of critical background references. It
was first proposed in a mechanistically consistent box-model from the temporal and spatial
characteristics of wind stress and Ekman transport at each of the 6 main upwelling centres by Monteiro
1996; 2010. As the authors suggest the Santana Casiano 2009 observations in the NBUS are beyond
their own boundaries of the upwelling system so the outgassing conclusion is derived mainly from
thermal impact on pCO₂.

*We thank the reviewer for this comment and have integrated the missing reference into the introduction*
*section of our manuscript.*

*"Considering that also model studies came to this conclusion, we see no issues related to resolving the*
*spatial and temporal variability and sampling biases, which we have addressed in our manuscript*
*(Lines 124-129) and spatio-temporal interpolation method."*

This comparison requires more than a cursory comment.

*We thank the reviewer for the comment. By including the spatio-temporal effect of the thermally*
*controlled pCO₂ and distinguishing between surface warming and biologically-mediated CO₂ uptake*
*effects, we have addressed the issue accordingly.*

*"However, what is new and what has been acknowledged by the reviewers is that we provided*
*additional data and followed a new notion to explain the opposing behavior of the two systems and the*
*BUS's role for the biological carbon pump by means of preformed nutrient utilization."*

New data is not equivalent to new insights. I do not see the point of including the observations based
pCO₂ and flux calculations, which are the outcome of both pre-formed and re-mineralized nutrient
fluxes, in a study that is primarily aiming to constrain the carbon export from pre-formed nitrate alone.
It's just an add on and it confuses the primary focus of the paper set out in Fig 4. This means that the
CO₂ fluxes and carbon export production fluxes are not integrated and that detracts from the
significance of the paper.

*The first part of our manuscript is relevant to understand the role of preformed nutrients for the regional*
*CO₂ sources/sinks, since not only the temperature effect is crucial for pCO₂, but also the use of*
*preformed nutrients (as already shown e.g. by Hales et al. 2005), which appears to have been*
*disregarded by reviewer #4. Hereby, however, it is important to also note that the use of preformed*
*nutrients plays a different role in the efficiency of the biological carbon pump from a broad-scale*
*perspective, as explained earlier in Issue 3.*

*"The underlying process-understanding that results from our study thereby leads to far reaching*
*conclusions regarding the role of Eastern Boundary Upwelling Systems as a CO₂ sink that balances*
*CO₂ losses at sites where preformed nutrients are formed, shedding new light on the BUS from a*
*budgeting and process perspective."*

As discussed below in (issue 3), this may be so but it requires a more careful discussion on the
limitations of this assertion.

*We have addressed this issue in the last comment to which we kindly refer.*

*"To issue 2: This can be divided into two parts, namely a) a discussion of earlier comparable studies in*
*the BUS and other Eastern Boundary Upwelling Systems, and b) a discussion of the differences*
*between CO₂ flux budgets and new production fluxes, both of which were considered in the*
*manuscripts pointed out in the following:*

*a) In view of earlier comparable studies in the BUS, both Monteiro (2009) and a follow up work by*
*Waldron et al. (2009) established a carbon budget for the BUS with the inclusion of new production*

rates, and compared carbon losses from the BUS with CO₂ invasion rates as derived from Santana-
Casiano et al. (2009). As for our study, we followed this approach and compared new production rates
obtained from Waldron et al. (2009), Emeis et al. (2018) and Monteiro (2009) with CO₂ fluxes as
derived from our dataset which includes data from Santana-Casiano et al. (2009) and extensive
recordings representative for the coastal and shelf areas along the continental margin off Namibia and
South Africa. However, in contrast to previous studies from the BUS, we included impacts of the
solubility pump and differentiated between new production driven the utilization of preformed and
regenerated nutrients.”

Both Monteiro 2010 and Monteiro model in Waldron et al., 2009 correct the pCO₂ for warming of
upwelled waters inshore and offshore.

Yes, this has been done, but what we meant to say was that both solubility pump effects and the
preformed nutrient share in new production were not previously taken into consideration when
discussing air-sea gas exchanges in the BUS.

“A similar approach to elucidate the impact of preformed nutrients on carbon fluxes had already been
applied to the Oregon upwelling region in the California Current System (Hales et al., 2005), as was
mentioned in the Introduction of our manuscript. Hence, all previously published data from the BUS on
pCO₂ characteristics and new production, as well as concepts developed from other EBUS that are of
relevance to assess carbon budgets and the role of preformed nutrients, were included into our study.”

What is missing is a more thorough discussion of the consequences of the assumptions and choices
made in this study relative to historical work. This should include a table of both CO₂ fluxes and carbon
export fluxes and nutrient boundary conditions from the different studies.

Throughout our manuscript, we considered different factors when comparing our work with previous
studies. These include the choice of offshore boundaries (see manuscript lines 90-100), choice of
surface areas and dissociation constants for calculating air-sea CO₂ fluxes (see manuscript lines 139-
154), as well as the approaches for estimating potential new production rates and their implication for
new production ranges for the BUS (see manuscript lines 236-238: “ Hereby, despite of the inherent
methodology that could be held responsible for the discrepancy in the estimated new production rates,
they provide lower and upper cases to assess the magnitude of the effect preformed nutrient
consumption may hold in the BUS.”).

“b) In our discussion on the difference between CO₂ flux budgets and new production fluxes (Lines
185- 196), we included impacts of the solubility pump and differentiated between new production driven
by the supply of regenerated and preformed nutrients. This is new in terms of the BUS and crucial
because of the different roles these nutrients play for the biological carbon pump, which enabled us to
explain the opposing behavior of the two subsystems.”

I am not convinced that the last sentence logically follows from the first

We thank the reviewer for the comment. By including the spatio-temporal effect of the thermally
controlled pCO₂ and distinguishing between surface warming and biologically-mediated CO₂ uptake
effects, this issue has been addressed accordingly.

“To issue 3: Since we applied the well-known influence of preformed nutrients on the CO₂ uptake of the
biological carbon pump and the central role deep water formation in the Southern Ocean plays for the
cycling of preformed nutrients to the BUS, omitting the Southern Ocean would have only led to
misinterpretations of the results derived from the BUS. For instance, we showed that the biological
carbon pump takes up CO₂ in the BUS via the utilization of preformed nutrients. Ignoring that the
biological carbon pump loses CO₂ during the formation of preformed nutrients in the Southern Ocean
would have led to the impression that the BUS acts as a net CO₂ uptake region, whereas in steady
state, the CO₂ uptake through the utilization of preformed nutrients balances the CO₂ loss caused

during their formation. The latter is thereby illustrated in Figure 4, showing the cycling of preformed
 nutrients and their contribution to new production in the BUS.“

Yes this study provides a focus on pre-formed nutrient boundary conditions to the BUS but it fails to
 adequately discuss the implications of this potentially interesting approach and its underlying
 assumptions for what is a complex system, both biogeochemically and physically. Fig 4 seems to
 suggest a simple link between carbon export linked to pre-formed nutrients and compensation of CO₂
 outgassing in the Southern Ocean. In reality there is a complex set of physics and biogeochemical
 feedbacks that are well recognized and published that most likely influence the impact of these
 assumptions on the magnitude of the carbon export fluxes. These include shelf deposition especially in
 the NBUS, both aerobic and anaerobic remineralization of the exported carbon, de-nitrification,
 nitrification. I would not expect the authors to come up with a detailed study of these factors but at least
 discuss the contribution and uncertainty levels that they make to re-balancing the CO₂ loss in the
 Southern Ocean. The rather large number for the contribution by the Benguela to offsetting outgassing
 of CO₂ in the Southern Ocean requires stronger justification and clearer uncertainties.

The main elements in Figure 5 (former Figure 4) display the link between the BUS and Southern Ocean
 through the transport pathway of water masses. Hereby, we added a descriptive text stating that the
 transport of the water masses are shaping the preformed nutrient concentrations of SACW and
 ESACW before they reach the BUS and are being upwelled into the surface region. With this, we draw
 attention to the occurrence of transformative processes along the transport pathway through the South
 Atlantic, which are, yet, not well known, as also stated previously by Reviewer #2 during the 1st peer
 review process of our manuscript. By addressing the large range of published new production rates in
 which our own estimates fall into, we intended to shed light on the uncertainty of the role of the BUS's
 biological carbon pump in offsetting the CO₂ flux of the Southern Ocean, while further mentioning other
 factors (e.g., climate change & anthropogenic pressures like fisheries) that could impact the BUS's
 biological carbon pump. We agree that the extent, to which preformed-based new production is
 exported and further e.g., transferred into sediments is dependent on various processes as mentioned
 by Reviewer #4. However, our intention was to outline potential new production rates to elucidate the
 role of preformed nutrients once they reappear in the surface region within the BUS after their
 subduction in the Southern Ocean.

Broecker, Wallace, Taro Takahashi, and Timothy Takahashi. "Sources of Flow Patterns of Deep Waters as
 Deduced from Potential Temperature, Salinity and Initial Phosphate Concentration." *Journal of*
 *Geophysical Research* 90 (01/01 1985): 6925-39.

Eppley, Richard W., and Bruce J. Peterson. "Particulate Organic Matter Flux and Planktonic New Production in
 the Deep Ocean." *Nature* 282 (1979 1979).

González-Dávila, Melchor, J. Magdalena Santana-Casiano, and Ivan R. Ucha. "Seasonal Variability of Fco₂ in
 the Angola-Benguela Region." *Eastern Boundary Upwelling Ecosystems: Integrative and Comparative*
 *Approaches* 83, no. 1 (December 1, 2009 2009): 124-33.

Hutchings, L., C.D. van der Lingen, L.J. Shannon, R.J.M. Crawford, H.M.S. Verheye, C.H. Bartholomae, A.K.
 van der Plas, *et al.* "The Benguela Current: An Ecosystem of Four Components." *Eastern Boundary*
 *Upwelling Ecosystems: Integrative and Comparative Approaches* 83, no. 1 (December 1, 2009 2009): 15-
 32.

Ito, Takamitsu, and Michael J. Follows. "Preformed Phosphate, Soft Tissue Pump and Atmospheric Co₂." *Journal*
 *of Marine Research* 63 (2005 2005): 813-39.

Monteiro, P. M. S. "Carbon Fluxes in the Benguela Upwelling System." In *Carbon and Nutrient Fluxes in the*
 *Continental Margins: A Global Synthesis*, edited by K.K. Liu, L. Atkinson, R. Quiñones and L. Talaue-
 McManus. Berlin: Springer, 2010.

- Pytkowicz, R. M., and D. R. Kester. "Oxygen and Phosphate as Indicators for the Deep Intermediate Waters in the
Northeast Pacific Ocean." *Deep Sea Research and Oceanographic Abstracts* 13, no. 3 (1966/06/01/
1966): 373-79.
- Rae, C. M. Duncombe. "A Demonstration of the Hydrographic Partition of the Benguela Upwelling Ecosystem at
26°40's." *African Journal of Marine Science* 27, no. 3 (2005/12/01 2005): 617-28.
- Santana-Casiano, J. Magdalena, Melchor González-Dávila, and Ivan R. Ucha. "Carbon Dioxide Fluxes in the
Benguela Upwelling System During Winter and Spring: A Comparison between 2005 and 2006." *Surface*
*Ocean CO2 Variability and Vulnerabilities* 56, no. 8 (April 1, 2009 2009): 533-41.
- Shillington, Frank, Chris Reason, Christopher Duncombe Rae, P. Florenchie, and Pierrick Penven. "Large Scale
Physical Variability of the Benguela Current Large Marine Ecosystem (Bclme)." *Large Marine*
*Ecosystems* (October 14, 2013 2013).
